# Extrapolative Continuous-time Bayesian Neural Network for Fast Training-free Test-time Adaptation

**Hengguan Huang**[1]* **Xiangming Gu**[1] **Hao Wang**[2] **Chang Xiao**[1] **Hongfu Liu**[1] **Ye Wang**[1]*
[1]National University of Singapore    [2]Rutgers University

## Abstract

Human intelligence has shown remarkably lower latency and higher precision than most AI systems when processing non-stationary streaming data in real-time. Numerous neuroscience studies suggest that such abilities may be driven by internal predictive modeling. In this paper, we explore the possibility of introducing such a mechanism in unsupervised domain adaptation (UDA) for handling non-stationary streaming data for real-time streaming applications. We propose to formulate internal predictive modeling as a continuous-time Bayesian filtering problem within a stochastic dynamical system context. Such a dynamical system describes the dynamics of model parameters of a UDA model evolving with non-stationary streaming data. Building on such a dynamical system, we then develop extrapolative continuous-time Bayesian neural networks (ECBNN [2]), which generalize existing Bayesian neural networks to represent temporal dynamics and allow us to extrapolate the distribution of model parameters before observing the incoming data, therefore effectively reducing the latency. Remarkably, our empirical results show that ECBNN is capable of continuously generating better distributions of model parameters along the time axis given historical data only, thereby achieving (1) training-free test-time adaptation with low latency, (2) gradually improved alignment between the source and target features and (3) gradually improved model performance over time during the real-time testing stage.

## 1 Introduction

Human consciousness requires building an internal predictive model that describes the external world [1, 2]. Numerous neuroscience studies have suggested that humans continually update their internal predictive model to represent the statistics of the fluctuating environments accurately [3, 4, 5, 6]. Internal predictive modeling is crucial for humans to perceive a continuous stream of sensory inputs (e.g. sound and sight) in real-time, due to constraints on neural transmission delays [7]. For example, in order to catch a fast-moving object, the internal predictive model is required to forecast its trajectory ahead to compensate for the response latency of neurons; in order to recognize words from lip movements at various view angles in real-time, again, the internal predictive model is needed to ensure that decision outcome is not delayed [8].

Such a mechanism allows humans to process non-stationary streaming sensory data with remarkably low latency and high precision by continuously and instantly generating internal predictive models. In contrast, most existing domain adaptation algorithms either require the collection of additional data from a large amount of increasingly more domains for training [9], or the use of testing streaming data for test-time fine-tuning [10, 11, 12, 13], which is impractical for real-time streaming applications.

---

*Correspondence to: Hengguan Huang <huang.hengguan@u.nus.edu>, Ye Wang <wangye@comp.nus.edu.sg>

[2]Our code will soon be available online: https://github.com/guxm2021/ECBNN

36th Conference on Neural Information Processing Systems (NeurIPS 2022).

This leads us to rethink why existing approaches in unsupervised domain adaptation (UDA) fail to achieve what humans are capable of when handling streaming data.

This work aims to introduce "internal predictive modeling" to augment traditional UDA algorithms in handling non-stationary streaming data. A UDA, augmented with such a mechanism, should be able to (1) instantly extrapolate the model parameters throughout the real-time testing stage instead of simply adapting the model with unacceptable latency and (2) align features from the source stream and the target stream globally rather than local data streams. However, the development of a mathematical framework that can encode such a mechanism involves several challenges: (1) Real-world non-stationary streaming data usually are highly uncertain in their temporal dynamics and therefore require us to design a model that can encode *uncertainty* of the temporal dynamics and produce *multimodal* distributions of model parameters. (2) Only partial observation of the time series or the local streams is available, resulting in poor alignment quality.

To tackle the first challenge, we propose formulating internal predictive modeling as a continuous-time Bayesian filtering problem within a stochastic dynamical system context. Such a system describes the dynamics of UDA's model parameters that are evolving with non-stationary streaming data. To our knowledge, such Bayesian treatment of time-evolving neural network (NN) parameters is mostly unexplored. Existing continuous-time Bayesian filtering approaches, e.g. CTBN [14], assume discrete-state for the dynamical system; and thus cannot scale to NNs. Furthermore, most of the Bayesian neural networks [15, 16] are designed to reason about static distributions of NN parameters. In this paper, we propose an extrapolative continuous-time Bayesian neural network (ECBNN), which generalizes existing static BNNs to encode uncertainty of the temporal dynamics and allows inferring multi-step ahead model distributions before observing the incoming data, therefore effectively reducing the latency. Furthermore, ECBNN adopts a weighted set of particles to represent the posterior distribution, which can represent multimodal distributions with arbitrary shapes using a sufficient number of particles.

To tackle the second challenge, we propose to perform UDA on the entire data generation mechanism, which is described by a latent stochastic process conditioned on historical data. This leads to another challenge: aligning two latent stochastic processes tends to result in an intractable discriminator loss. We further derive an analytical upper bound for the discriminator loss. Calculating such a bound requires evaluating posterior distributions of the UDA encoder in continuous time; this then leads us to reformulate the problem by deriving a new particle filter differential equation (PFDE) rather than simply applying traditional Bayesian filtering methods, such as particle filter (PF) [17]. It is worth noting that we adopt an efficient implementation for solving the PFDE. Specifically, we adopt surrogate neural networks to approximate the solution. Therefore, the time-evolving model distributions can be efficiently inferred by a single forward pass of the associated surrogate neural networks.

We list our major contributions as follows:

(i) To the best of our knowledge, this work is the first to explore how to incorporate the mechanism of "internal predictive modeling" into traditional UDA algorithms in handling non-stationary streaming data for real-time applications with low latency requirements.

(ii) We present the first Bayesian treatment of time-evolving NN parameters for UDA with streaming data and propose extrapolative continuous-time Bayesian neural network (ECBNN).

(iii) We derive a novel particle-filter-based differential equation, thereby providing an analytical upper bound for ECBNN to achieve temporal-domain-invariant representation learning.

(iv) We empirically demonstrate that our approach is able to achieve (1) training-free test-time adaptation with low latency, (2) gradually improved alignment between the source and target features and (3) gradually improved model performance over time during the real-time testing stage.

## 2   Related Work

**Brain-informed Artificial Intelligence.**    In recent years, artificial intelligence (AI) has seen tremendous advances and breakthroughs in building intelligent systems that display human perceptual and cognitive abilities, including dialogue system [18], automatic speech recognition [19, 20] and

high-fidelity image generation [21], among others. However, mainstream AI systems have not yet reached the capabilities of a highly sophisticated bio-intelligence. Human brains are the only evidence that general intelligence at the human level is possible. Brain-inspired AI [22], aiming to bridge the gap between artificial intelligence and brain science, play a vital role in opening up the field of artificial neural network and continuing to provide the foundation for research on deep learning and reinforcement learning [23]. However, such an AI framework only takes modern theories in brain science as rough guidance for developing new AI algorithms, and it mostly ignores biological plausibility. [24, 25] proposed a brain-informed AI framework, which consists of a task-specific component for achieving a variety of AI tasks, and a brain-informed component for encoding a specific brain science mechanism. For instance, deep graph random process (DGP) [24] incorporates relational thinking, an assumption of cognitive neuroscience, to conversational AI, which enables the learning of the relational structure from the conversations without requiring any relational labels during training; stochastic boundary ordinary differential equation (STRODE) [25] links the time perception and postdiction mechanism with generative AI and conversational AI respectively, allowing directly learning event-times from time-series and creating a neurally plausible model whose outputs match the postdiction assumption. This work can also be classified as a brain-informed AI, where the internal predictive modeling is integrated into unsupervised domain adaptation, enabling the model to learn continuously and immediately from the non-stationary streaming data during real-time testing.

**Unsupervised Domain Adaptation with Streaming Data.** There is a vast body of prior work [26, 27, 28, 29, 30, 31] exploring domain-invariant representation learning for unsupervised domain adaptation (UDA). These works' key idea is harnessing the expressive power of neural networks to transform the input data into latent representations that are further aligned across static domains. However, in real-time streaming applications, streaming data usually follows non-stationary distributions. Consequently, these work and a few other test-time adaptation methods [12, 32, 13] mostly fail to handle such data due to the assumption of static domains or the requirements of low latency.

The UDA for non-stationary streaming data has been studied before but in different problem settings with different focuses. For example, continuous manifold adaption (CMA) [9] addresses this problem by adapting the model with the additional unlabeled data from several pre-defined intermediate domains; predictive domain adaptations [33, 10] attempt to solve a specific scenario where target data is unavailable but the metadata that characterizes the target domain is provided. Continuously indexed domain adaptation [30] addresses this problem by jointly adapting across continuously indexed domains with domain indices such as 'time' and other domain attributes. However, these methods cannot be directly applied to our problem settings due to the unavailability of the meta-data, e.g. absolute time stamps of each data point. The closest work to ours is [11], in which an online test-time adaptation framework is introduced to handle streaming data with evolving domains. However, [11] requires using partial test data for test-time fine-tuning, greatly limiting the applicability to real-time streaming situations. In contrast, our ECBNN allows reasoning multi-step ahead posterior distribution of UDA's model parameters before observing the incoming data, thereby achieving training-free test-time adaptation with much lower latency.

**Bayesian Neural Networks with Dynamical Systems.** Bayesian neural networks (BNNs) [34] aim to infer the posterior distribution of the NN parameters given observations and a prior imposed on these parameters. With a different aim, the early work on neural dynamical systems, e.g. neural ordinary differential equation (ODE) [35], is aimed at approximating the dynamics of layer-wise representations of deep NNs by using a non-Bayesian NN. Bayesian Neural ODEs [36, 37] seek to improve neural ODE by replacing the non-Bayesian NN with a BNN. Unlike the above studies, our work aims to develop a generalized BNN that captures the time-evolving posterior distribution of the NN parameters over time. A different strand of research focuses on building a continuous-depth BNN using stochastic differential equations [15]. Although it is close to our work, their model assumes a static distribution for NN parameters; and thus cannot scale to capture the time-varying distribution of NN parameters. In general, our work belongs to the category of Bayesian deep learning (BDL) [38, 39, 24, 25, 40], and is the first BDL method that handles time-evolving NN parameters. (Please refer to Appendix E for related work on particle filtering with neural networks.)

## 3 Method

In this section, we first formalize the problem of test-time streaming domain adaptation for non-stationary streaming data, and then describe our methods for addressing the problem.

### 3.1 Problem Setting: Test-time Streaming Domain Adaptation

**Problem.** We consider the unsupervised test-time domain adaptation setting for non-stationary streaming data. Specifically, given a source stream $\{\mathbf{B}_i^s\}_{i=1}^I$, a target stream $\{\mathbf{B}_i^a\}_{i=1}^I$ and a testing stream $\{\mathbf{B}_i^e\}_{i=1}^I$, where $I$ denotes the number of data batches; the source data batches $\mathbf{B}_i^s = \{\mathbf{x}_t^s, \mathbf{y}_t^s\}_{t=(i-1)\cdot N}^{i\cdot N-1}$ are labeled, while the target and testing data batches $\mathbf{B}_i^a = \{\mathbf{x}_t^a\}_{t=(i-1)\cdot N}^{i\cdot N-1}$, $\mathbf{B}_i^e = \{\mathbf{x}_t^e\}_{t=(i-1)\cdot N}^{i\cdot N-1}$ are unlabeled, where $N$, the number of frames in each batch, is usually small for machine learning applications that require streaming inputs, e.g. streaming video processing systems. We first train a UDA model based on the source and target streams $\{\mathbf{B}_i^s\}_{i=1}^I$ and $\{\mathbf{B}_i^a\}_{i=1}^I$; with the trained model, we aim to perform adaptation as each testing stream data batch $\mathbf{B}_i^e$ arrives.

**Low Latency and Timeline.** Notably, during the real-time testing stage, our problem setting has strict requirements for the latency caused by adaptation. Therefore, existing offline adaptation methods or online adaptation methods that require training or fine-tuning are not applicable to our problem setting. Furthermore, the source stream, target stream, and testing stream may not necessarily share the same timeline, and the absolute time stamps of the data are unknown; thus, DA methods [30] that require absolute time stamps fail to handle our problem setting as well.

### 3.2 Overview

Below we provide an overview of our method (Fig. 1).

**Training.** Our approach adopts both source stream and target stream data for training: (i) it starts by adopting PFDE, taking as input the historical streaming batches $\mathbf{B}_{1:i-1}$, to infer the posterior distributions of NN parameters of the encoder $f_e$; based on such distributions, particles are sampled to update the encoder $f_e$ (updated before processing $\mathbf{B}_i$); (ii) it further takes the updated encoder to generate $z$, the encoder output for $\mathbf{B}_i$, and $\widetilde{z}$, the encoder output for the approximated data required for calculating the proposed analytical upper bound (see Sec.3.6 for more details of the approximated data); (iii) finally, we follow the typical adversarial UDA training procedure to evaluate the training loss given $z$ and $\widetilde{z}$ and update the whole framework (see Sec. 3.5 for more detailed description).

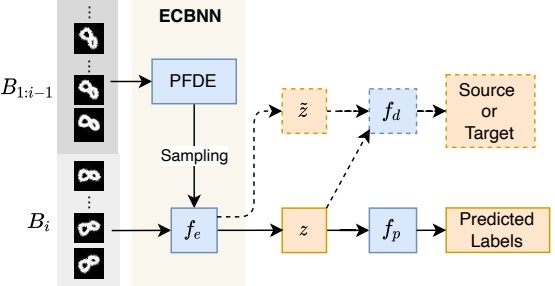

Figure 1: General framework of real-time streaming domain adaptation; the components marked with a dotted line are removed during testing.

**Real-time Testing** During the real-time testing stage, the testing stream cannot be revisited and should be processed as each data batch arrives. The testing procedure is mostly similar to the training, except that discriminator are removed.

### 3.3 Particle Filter Differential Equation (PFDE)

Traditional PF methods (see Appendix D on Background of Particle Filtering), e.g. bootstrap filters [17], require calculating likelihood for observations at each time point, which is infeasible in our problem setting since only historical data is available when performing multi-step ahead Bayesian inference during the real-time testing stage. We therefore develop, by the following theorem, a new type of dynamical system, dubbed Particle Filter Differential Equation (PFDE), that describes the evolution of the state in an input-driven continuous-time stochastic dynamical system.

**Theorem 1.** *Suppose we are given a time series* $\{\mathbf{x}_t\}_{t=0}^{+\infty}$. *Assume* $\{\mathbf{x}_t\}_{t=0}^{+\infty}$ *are conditionally independent given a sequence of latent states* $\{\theta(t)\}_{t=0}^{+\infty}$, *which is itself assumed to be Markovian. Assume marginal distribution of the state* $p(\theta(t))$ *is stationary. Let* $\tau \in [0, \epsilon]$, *where* $\epsilon$ *is small positive constant. Assume* $\mathbf{x}_s$ *with* $s \in [s - \epsilon, s + \epsilon]$ *is stationary. Let* $p(\theta(t)|\theta(t - \tau))$ *be the transition distribution. Let* $w(t)^{(j)}$ *be the importance weight. Then we have:*

$$\left(\log w(t)^{(j)}\right)'' = \left(\log p(\theta(t)^{(j)}|\theta(t - \tau)^{(j)})\right)'',$$

*where $(\cdot)'$ and $(\cdot)''$ denote the first-order and second-order temporal derivatives respectively.*

The derivation of PFDE in Theorem 1 (see Appendix A for the proof) is based on the bootstrap filter. By assuming the marginal distribution of the state $p(\theta(t))$ as a stationary distribution (which is a mild assumption as conditional distribution can still be non-stationary), we successfully build the *link between the transition distribution and the importance weight*, whose calculation no longer requires calculating the intractable likelihood term.

### 3.4 Extrapolative Continuous-time Bayesian Neural Networks (ECBNN)

At ECBNN's core is a PFDE, which provides a Bayesian treatment of time-evolving NN parameters and inherits several advantages from both particle filters and differential equations. For instance, PFDE enables inferring posteriors of NN parameters along the time axis by solving itself at any given time point.

Specifically, due to the intractable parametric form of the transition distribution $p(\theta_t|\theta_{t-1})$ in our setting, we use an approximate distribution $\widetilde{q}(\theta(t)) = \mathcal{N}(m(t), s(t))$ as the importance distribution, where $m(t)$ and $s(t)$ are the mean and variance of the Gaussian. Let us denote $w(t)^{(j)}$ as the importance weight and $\theta(t)^{(j)}$ as the particle. By continuous-time Markovian assumption and Theorem 1, the PFDE can be written as:

$$f(m(t), m'(t), m''(t), s(t), s'(t), s''(t), \theta(t)^{(j)}) = (\log(w(t)^{(j)})'', \tag{1}$$

where $f(\cdot)$ is a nonlinear function, whose detailed form is described in Appendix G. It follows that the solution of PFDE can be written as:

$$G(t) = \{m(t), s(t), \{w(t)^{(j)}\}_{j=1}^{k}\}. \tag{2}$$

Then ECBNN takes the following form to approximate the posterior of the NN parameters $q(\theta(t))$:

$$q(\theta(t)) = \sum\nolimits_{j=1}^{k} w(t)^{(j)}\delta(\theta(t) - \theta(t)^{(j)}), \quad \theta(t)^{(j)} \sim \widetilde{q}(\theta(t)) = \mathcal{N}(m(t), s(t)), \tag{3}$$

where $\delta$ denotes the Dirac-delta function. We further demonstrate the inference and learning of ECBNN by applying it to address the problem of test-time streaming DA.

### 3.5 Application of ECBNN for Test-time Streaming Domain Adaptation

This section demonstrates how to embed ECBNN into a typical UDA framework for test-time streaming DA.

**Learning.** A UDA usually contains an encoder $f_e$ to project the input to a domain-invariant space $\mathbf{z}$ with the help of a discriminator $f_d$, such that the target data can be accurately predicted by the predictor $f_p$. Denoting $\theta$ as the parameters of the encoder $f_e$, a UDA performs a minimax optimization with the value function $V(f_e, f_d, f_p)$ as:

$$\min_{f_e(\cdot;\theta),f_p} \max_{f_d} V_p(f_e(\cdot;\theta), f_p) - \lambda_d V_d(f_d, f_e(\cdot;\theta)), \tag{4}$$

where $V_p$ is the expectation of the prediction loss (e.g., cross-entropy loss for classification tasks) over the source data batches; $\lambda_d$ is a hyperparameter balancing both losses ;$V_d$ is the expectation of the discriminator loss over source and target data batches. For simplicity, we only emphasize the parameters of the encoder $f_e$ and omit other parameters.

We extend a UDA with ECBNN to handle test-time streaming DA by adopting ECBNN as the encoder of UDA. The architecture of our model is shown in Fig. 1. Recall that ECBNN adopts an approximate distribution $\widetilde{q}(\theta(t))$ as the importance distribution due to the intractability of the transition distribution. This leads us to adopt variational inference to optimize our model, in which $\widetilde{q}(\theta(t))$ is treated as the variational distribution of the transition distribution. Let us denote $p(\theta(t))$ as the prior and $l_u(\cdot;\theta, f_p, f_d)$ as the function inside the minimax optimization of Eq.(4). Since both terms in $l_u$ can be treated as negative log-likelihood, we directly combine their negation's expectation with KL terms as the evidence lower bound (ELBO), which can be written as:

$$l_b = \sum\nolimits_{t=1}^{I \times N} (-\mathrm{KL}(\widetilde{q}(\theta(t))||p(\theta(t))) - \sum\nolimits_{j=1}^{k} w(t)^{(j)} l_u(\cdot;\theta(t)^{(j)}, f_p, f_d). \tag{5}$$

In the following subsections, we will demonstrate how to obtain the solution of PFDE and how we can jointly learn ECBNN and UDA under the constraints of the PFDE.

**Inference.** The key to reducing latency is to force the inference of variables involved in PFDE to depend on historical batch data only. Specifically, suppose we are given up to $i-1$-th batch $\mathbf{B}_{1:i-1}$; we aim to infer the encoder distribution for each frame of the incoming batch by solving the PFDE. However, existing ODE numerical solvers fail to give the approximate solution as the initial or boundary values of the variables are computationally intractable. Inspired by [25, 41], we adopt surrogate neural networks to approximate the solution of these variables, such that:

$$m(t) = f_{\theta_m}(t, \mathbf{B}_{1:i-1}), \quad s(t) = f_{\theta_s}(t, \mathbf{B}_{1:i-1}), \quad \log w(t)^{(j)} = f_{\theta_w}(t, \mathbf{B}_{1:i-1}, \theta(t)^{(j)}), \quad (6)$$

where $f_{\theta_*}$ are surrogate neural networks ( implementations detail in Appendix G). To encourage such solutions to satisfy the governing differential equation of PFDE defined in Eq. (1), we further include in the ELBO the following additional loss term $l_{de}$:

$$l_{de} = \sum_{t=1}^{I \times N} |f(m(t), m'(t), m''(t), s(t), s'(t), s''(t), \theta(t)^{(j)}) - (\log(w(t)^{(j)}))''|^2. \quad (7)$$

To process $\mathbf{B}_{1:i-1}$, $f_{\theta_*}$ contains a recurrent neural network (RNN), which maintains hidden states to represent the historical batches without the need to repeatedly process the sequence from the start. Notably, our solution doesn't require absolute time stamps, e.g. to solve for the encoder distribution for the incoming batch at the first frame, we can set $t = 1$. Furthermore, we specifically design these neural networks as a non-linear function of time $t$, allowing all temporal derivatives required by Eq. (7) to be computed by the chain rule and evaluated by means of automatic differentiation [42].

**Prior.** We use NNs to parameterize the mean $m_0(t)$ and variance $s_0(t)$ of the prior $p(\theta(t))$:

$$m_0(t) = f_{\theta_{m0}}(\mathbf{B}_i), \quad s_0(t) = f_{\theta_{s0}}(\mathbf{B}_i), \quad (8)$$

where $f_{\theta_*}$ are neural networks (implementations detail in Appendix G). Recall that we force the posterior to depend on historical data batches only. To encourage the posterior to *extrapolate the future*, we take a further step and adopt the current batch to estimate the prior for "future parameters". It's worth noting that this will not affect the latency as the prior will be removed during testing.

### 3.6 Toward Temporal-domain-invariance for Streaming Data

Due to the drifting nature of the non-stationary streaming data, aligning features extracted across the source and target streams is critical yet challenging for the success of unsupervised domain adaptation (UDA). Simply aligning using partial observation of the time series leads to poor alignment quality. To address this challenge, we proposed to perform UDA on the entire data generation mechanism. Similar to particle filters, PFDE is also capable of capturing the latent time-varying distribution of the non-stationary streaming data. Therefore, the solution of PFDE $G(t) = \{m(t), s(t), \{w(t)^{(j)}\}_{j=1}^k\}_0^{+\infty}$ can be treated as a stochastic process that describes the generation mechanism of the non-stationary data. We, therefore, give a temporal-domain-invariant loss that is the expectation of the discriminator loss $V_d$ over the stochastic process $G(t)$:

$$l_i = \int_0^{+\infty} \sum_{j=1}^k w(t)^{(j)} V_d(f_d, f_e(\mathbf{x}(t); \theta(t)^{(j)})) dt, \quad (9)$$

However, the upper limit of integration in the above loss function approaching infinity results in an improper integral, which may not converge. To overcome the above challenges, we propose the following theorem, which provides an analytical upper bound for the temporal-domain-invariant loss.

**Theorem 2.** *Suppose we are given an arbitrary function, $\pi(t)$ with $t \in [0, +\infty)$. Let $\beta = -e^{-t^2/\alpha^2}$. Let $\epsilon$ be a positive real constant and let $h : [-1, 0) \to R$ be a continuous function. There exists an $H : [-1, 0) \to [0, +\infty)$ that satisfies an initial value problem:*

$$H'(\beta) = h(\beta), \quad H(-1) = 0, \text{ where } h(\beta) = \frac{-\alpha}{2\beta\sqrt{-\log(-\beta)}} \pi(\alpha\sqrt{-\log(-\beta)})$$

*Such that as $\epsilon \to 0$, we have:*

$$\int_0^{+\infty} \pi(t)dt \leqslant \lim_{\epsilon \to 0}(H(-\epsilon) + \|H(-2\epsilon) - H(-\epsilon)\|)$$

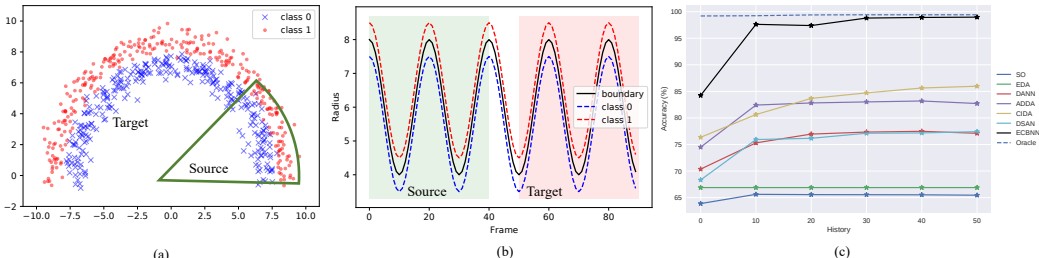

(a)  (b)  (c)

Figure 2: (a) Initial data samples at 0-th frame, where red dots and blue crosses are data samples from class 0 and class 1 respectively; Green sector separates the data on a circle into the source and target domain.. (b) Visualization of motion dynamics of decision boundary and data, where Black solid line represents the decision boundary, the red hash line represents data for class 0 and blue hash line for class 1, time ranges of source and target domain are indicated by different colors. (c) The accuracy of different methods by varying the length of historical stream data on Growing Circle testing set.

Let $\pi(t)$ be the integrand of the right-hand-side of the Eq. (9), Theorem 2 (proof given in Appendix B) states that we can construct a $H : [-1, 0) \to [0, +\infty)$ that satisfies an initial value problem (IVP):

$$H'(\beta) = h(\beta), \quad H(-1) = 0$$

where $h(\beta)$ can be written as:

$$h(\beta) = \frac{-\alpha}{2\beta\sqrt{-\log(-\beta)}} \sum_{j=1}^{k} [w(\phi)^{(j)} V_d(f_d, f_e(\widetilde{\mathbf{x}}(\phi); \theta(\phi)^{(j)}))], \tag{10}$$

where $\phi = \sqrt{-\log(-\beta)}$; $\widetilde{\mathbf{x}}(\phi)$ is the approximated data. Note that we use $\widetilde{\mathbf{z}}$ to denote the encoder output for the approximated data. Then by Theorem 2, the upper bound of the temporal-domain-invariant loss, $\widetilde{l}_i$, can be written as:

$$l_i \leqslant \widetilde{l}_i = \lim_{\epsilon \to 0} (H(-\epsilon) + \|H(-2\epsilon) - H(-\epsilon)\|), \tag{11}$$

where $H(-\epsilon)$ and $H(-2\epsilon)$ can be obtained by solving the IVP. For instance, these two terms can be approximated using the ODE numerical solvers such as the Euler method and the Runge-Kutta methods [35]. In this work, we adopt the Euler method to calculate these two terms, in which $\epsilon$ is set as the step size of the Euler method. Notably, by setting a suitable step size, $\widetilde{\mathbf{x}}(\phi)$, where $\phi \in (0, N)$ is required to calculate the bound; and thus $\widetilde{\mathbf{x}}(\phi)$ can be approximated by simply using linear interpolation over the hidden representation of the current batch. In summary, the final loss of our model can be written as a minimax optimization problem:

$$\min_{f_\theta, f_{\theta_0}, f_p} \max_{f_d} \left( -l_b(\cdot; f_\theta, f_{\theta_0}, f_p, f_d) + \lambda_{de} l_{de}(\cdot; f_\theta) - \lambda_i \widetilde{l}_i(\cdot; f_d, f_\theta) \right), \tag{12}$$

where $\lambda_{de}$ and $\lambda_i$ are importance weights; $f_\theta = \{f_{\theta_m}, f_{\theta_s}, f_{\theta_w}\}$, and $f_{\theta_0} = \{f_{\theta_{m0}}, f_{\theta_{s0}}\}$. (See Appendix C for the detailed form of training loss.)

## 4 Experiments

We evaluate ECBNN on both synthetic and real-world streaming data. Specifically, we conduct preliminary experiments on low dimensional synthetic toy data (Growing Circle), and then study the generalization of ECBNN in handling high dimensional synthetic video dataset based on handwritten digits (Streaming Rotating MNIST→USPS). We finally evaluate our model on OuluVS2, a realistic multi-view lip reading dataset. (See Appendix G for the detailed data configuration of datasets)

### 4.1 Baselines

We compare ECBNN with state-of-the-art UDA methods including **DANN** [27], **ADDA** [28], **CIDA** [30], and **DSAN** [29]. Since these UDAs assume static domains, we extend them to handle streaming data in our experiments by adopting a recurrent neural network (RNN) in the encoder. Since

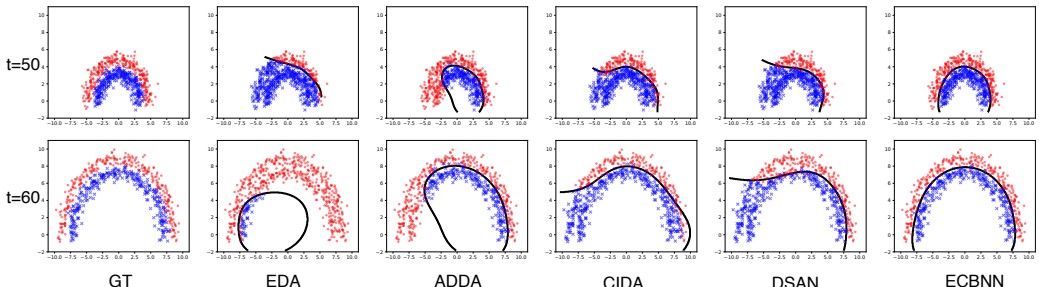

Figure 3: Results on the Growing Circle testing set. Black lines represent the decision boundaries generated according to model predictions.

absolute time stamps are unknown, we extend CIDA by using both the frame index and categorical domain label as the input to the encoder. Notice that all the above methods can only perform offline adaptation, while ECBNN can perform online test-time adaptation by continuously producing encoder distributions. We also compare our model with a typical online UDA method, (**EDA**) [11], which is able to perform online test-time adaptation but requires collecting data batches for fine-tuning. For fair comparisons, all models have similar numbers of weight parameters, with a similar neural network architecture of encoder $f_c$, predictor $f_p$, and discriminator $f_d$. Following [30], $\lambda_d$ is chosen from $\{0.2, 0.5, 1.0, 2.0, 5.0\}$ (see implementations details in Appendix G).

## 4.2 Toy Dataset

**Growing Circle.** We conduct a preliminary study on a toy dataset, Growing Circle, for a binary classification task; Growing Circle extends the Circle dataset in [30] to incorporate temporal changes. Fig. 2 shows the initial data samples at the 0-th frame, in which a green sector separates the data on a circle into the source and target domain. Fig. 2 (b) shows the motion dynamics of decision boundary and data, in which time ranges of the source and target domain are indicated by different colors. We include 0-th to 39-th frame to source domain and 50-th to 89-th to target domain.

**ECBNN Can Capture Time-evolving Decision Boundaries.** We visualize the predicted classification boundaries generated by different methods at frames $t = 50$ and $t = 60$, shown in Fig. 3 (more results in the Appendix H). We can see that ECBNN almost captures the ground truth (GT) decision boundaries at different time stamps, while other methods perform poorly when handling data with time-varying distributions.

**ECBNN Can Effectively 'Learn' from Testing data.** To evaluate ECBNN's capability of continuously generating better distribution of model parameters from historical stream data in real-time, we test our model by varying the length of the historical stream data ranging from 0 to 50 frames on the testing set. Fig. 2(c) shows the results for ECBNN and baselines. ECBNN consistently outperforms baselines. Interestingly, ECBNN's performance grows increasingly better with the accumulation of observed history, demonstrating ECBNN's potential in continuously learning from streaming data during real-time testing.

## 4.3 Streaming Rotating MNIST→USPS

**Streaming Rotating MNIST→USPS.** We further evaluate ECBNN on high-dimensional streaming data. To do this, we construct a synthetic video dataset based on MNIST and USPS: Streaming Rotating MNIST→USPS. The video sequences are generated from the rotating MNIST and USPS handwritten digits with sinusoidal angular velocity. We take the first 40 frames of the Streaming Rotationg MNIST video sequence as source data, whose rotating angles range from $0°$ to $80°$; we take the 50-th to 90-th frames of the Streaming Rotationg USPS video sequence as target data, whose rotating angles range from $110°$ to $190°$; we also take the 90-th to 110-th frames of the Streaming Rotationg USPS video sequence to construct an additional Out-of-Domain (OOD) testing set. (See Appendix G for the detailed data configuration)

**ECBNN Can be Generalized to High-dimensional Data.** We report the frame-level classification accuracy on the source-testing set, the target-testing set and the OOD-testing set from the source domain, the target domain and out-of-domain, respectively, as shown in Table 1.

We can see that Source-Only achieves an accuracy of 28.8%, which indicates this task is very challenging. ECBNN achieves the best accuracy of 60.9%. It dramatically outperforms other methods, e.g. it outperforms the best-performing baseline method (ADDA) by 8.9% absolute.

For domain generalization, ECBNN outperforms the best-performing baselines by 4.2% absolute, demonstrating ECBNN's potential of temporal-domain-invariant representation learning in the entire timeline.

We further evaluate ECBNN's ability to continuously generate better distribution of model parameters from high-dimensional stream data in the target domain. We evaluate ECBNN and baselines on the Rotating USPS testing set and report the accuracy at each frame in Fig. 4. Remarkably, while handling the real-time stream, ECBNN achieves gradually better performance over time. This demonstrates that ECBNN can continuously generate better encoder distribution over time based on historical data, whereby it becomes better and better at handling the data from the target domain.

**Ablation Study.** Our final loss has three parts, namely the ELBO, $l_b$, PFDE loss, $l_{de}$, and the upper bound of the temporal-domain-invariant loss, $\widetilde{l}_i$. To verify the effectiveness of each part, we conduct an ablation study on the Streaming Rotating MNIST→USPS dataset. Specifically, we design two ablation experiments: (i) removing $\widetilde{l}_i$ from the final loss; (ii) removing both $\widetilde{l}_i$ and $l_{de}$ from the final loss. Table 2 gives the frame-level classification accuracy on the target-testing set and the out-of-domain-testing set (OOD-testing set), from the target domain and "out-of-domain", respectively. We can see that by removing the upper bound of the temporal-domain-invariant loss, $\widetilde{l}_i$, the accuracy drops dra-

Table 1: **Accuracy (%) for various DA methods on Streaming Rotating MNIST→USPS Dataset**. We report the accuracy at the source domain, target domain and out-of-domain.

| Method | MNIST | USPS | |
| --- | --- | --- | --- |
| | Source | Target | OOD |
| Source-Only | 97.8 | 28.8 | 24.9 |
| DANN [27] | 97.7 | 45.9 | 33.0 |
| ADDA [28] | 97.3 | 52.0 | 34.3 |
| CIDA [30] | 97.5 | 46.5 | 31.1 |
| DSAN [29] | 96.1 | 46.8 | 33.0 |
| EDA [11] | **97.9** | 45.5 | 31.9 |
| ECBNN (Ours) | 97.1 | **60.9** | **38.5** |

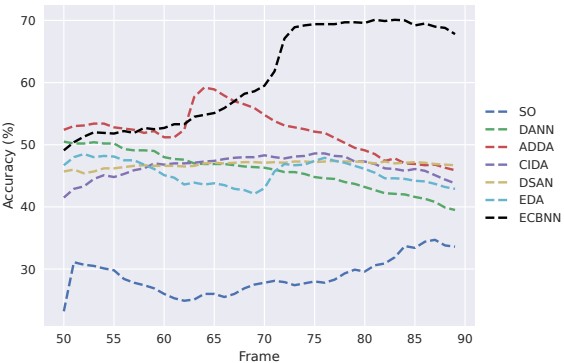

Figure 4: The frame-wise accuracy of different methods on Streaming Rotating USPS testing set.

matically for both the target domain and out-of-domain by 7.3% and 8.6%, respectively. The accuracy drop is more severe for "out-of-domain", verifying the effectiveness of the proposed upper bound of the temporal-domain-invariant loss in generalizing to unseen streaming data. Moreover, the accuracy drops further for both target domain and out-of-domain after removing the PFDE loss, $l_{de}$; this verifies the effectiveness of the PFDE loss.

Table 2: Ablation study results ( accuracy (%) ) on Streaming Rotating MNIST→USPS dataset

| Method | Target Domain ↑ | Out-of-Domain ↑ |
| --- | --- | --- |
| ECBNN (ours) | 60.9 | 38.5 |
| w/o $\widetilde{l}_i$ | 53.6 (-7.3) | 29.9 (-8.6) |
| w/o $\widetilde{l}_i$ & w/o $l_{de}$ | 43.2 (-17.5) | 28.0 (-9.5) |

## 4.4 Multi-veiw Lip Reading

**OuluVS2.** OuluVS2 dataset was originally proposed by [43], which is, as far as we know, the only publicly available dataset that contains multi-view lip motion data. It includes lipreading data recorded by 52 speakers from 5 different views. The train, valid and test sets include 5250, 750 and 1800 sequences, respectively. We adopt experiment settings similar to [44], e.g. the sequence-level

class label is voted by class labels of all frames. To formulate the streaming domain adaptation scenes, we consider the sequences from $0°$ view as the source domain and sequences from $[30°, 45°, 60°, 90°]$ as target domains. Besides, each data batch contains ten frames, and each frame is a greyscale image of size $44 \times 50$.

**ECBNN Can be Generalized to Real-world Streaming Data.** Table 3 shows the sequence-level classification accuracy of the baseline models and our new ECBNN on the Test of OuluVS2. We can see that ECBNN performs the best among all models in terms of accuracy in both source and target domains. ECBNN outperforms EDA, the online adaptation baseline, by $5.0\%$ absolute, while having remarkably lower latency in generating model parameters. We also report training time per epoch and the detailed accuracy of various DA methods on different views in Appendix G.

**ECBNN Can Produce Gradually Improved Alignment.** The production of lip motion data is governed by highly nonlinear dynamics over time, e.g. the movements of various articulators. Therefore, aligning such data in terms of the highly nonlinear temporal domain

Table 3: **Run time (s) and Accuracy (%) for various DA methods on OuluVS2 Dataset** We report run time per batch necessary to generate NN parameters and the sequence-level accuracy at the target domain.

| Method | Run time | Target |
|---|---|---|
| Source-Only | - | 46.0 |
| DANN [27] | - | 71.5 |
| ADDA [28] | - | 69.3 |
| CIDA [30] | - | 63.0 |
| DSAN [29] | - | 66.4 |
| EDA [11] | 0.23 | 70.3 |
| ECBNN (Ours) | **0.01** | **75.3** |

dynamics is very challenging. Here, we aim to qualitatively evaluate whether ECBNN can produce encoder distributions that achieve temporal-domain-invariant feature learning for real data. We randomly select data from 4 classes for both the source domain ($0°$) and the target domain ($30°$), and use t-SNE to visualize the corresponding encoder output $\mathbf{z}$ at time $t = 9, 15,$ and $21$. Fig. 5 shows the visualization. Surprisingly, the alignment quality increases over time, and features at $t = 21$ align much better than features at $t = 9$. This suggests that ECBNN can gradually improve the alignment given sequentially arriving data streams, thereby gradually improving accuracy on the target streaming data over time.

## 5 Conclusion

This paper extends the brain-informed artificial intelligence family with internal predictive modeling. Building on the connection between Bayesian neural networks and continuous-time stochastic dynamical system, we derive a novel particle-filter-based differential equation, thereby proposing extrapolative continuous-time Bayesian neural network (ECBNN), which generalizes existing Bayesian neural networks to represent temporal dynamics. We then augment the unsupervised do-

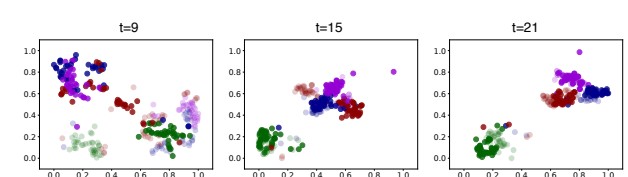

Figure 5: t-SNE visualizations of features at source domain (in *light* colors) and target domain (in *dark* colors) sampled at time $t = 9, 15,$ and $21$. We use different colors to mark different classes.

main adaptation framework by adopting ECBNN as its encoder for handling non-stationary streaming data. We further provide an upper bound with a theoretical guarantee for ECBNN to achieve temporal-domain-invariant representation learning. We show that ECBNN gradually improves model performance over time with low latency during the real-time testing stage. Our current approach assumes continuous dynamics of NN parameters; we leave it for future work to be able to learn more complex discrete dynamics introduced by discrete events.

## Acknowledgments

The authors would like to thank the anonymous reviewers for their insightful comments and suggestions and Dr. Qin Ling for her assistance in proofreading the initial manuscript. This project was partially funded by research grant R-252-000-B78-114. HW is partially supported by NSF Grant IIS-2127918.

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
