# A   Proof of Theorem 1

*Proof.* By definition of bootstrap filter, we have:

$$w(t)^{(j)} \propto w(t-\tau)^{(j)} p(\mathbf{x}_t | \theta(t)^{(j)})$$

$$\propto w(t-\tau)^{(j)} \frac{p(\theta(t)^{(j)} | \mathbf{x}_t) p(\mathbf{x}_t)}{p(\theta(t)^{(j)})}$$

$$\propto w(t-\tau)^{(j)} \frac{p(\theta(t)^{(j)} | \mathbf{x}_t)}{p(\theta(t)^{(j)})}$$

Given that $\mathbf{x}_s$ with $s \in [s-\epsilon, s+\epsilon)$ is stationary. Then there exists a constant $B$, such that:

$$w(t)^{(j)} = B w(t-\tau)^{(j)} \frac{p(\theta(t)^{(j)} | \mathbf{x}_t)}{p(\theta(t)^{(j)})} \tag{13}$$

$$= B w(t-\tau)^{(j)} \frac{p(\theta(t)^{(j)} | \mathbf{x}_{t-\tau})}{p(\theta(t)^{(j)})}$$

$$= B w(t-\tau)^{(j)} \frac{p(\theta(t)^{(j)} | \theta(t-\tau)^{(j)}) p(\theta(t-\tau)^{(j)} | \mathbf{x}_{t-\tau})}{p(\theta(t)^{(j)})} \tag{14}$$

Delay Eq. (13) by $\tau$, we have:

$$w(t-\tau)^{(j)} = w(t-2\tau)^{(j)} \frac{p(\theta(t-\tau)^{(j)} | \mathbf{x}_{t-\tau})}{p(\theta(t-\tau)^{(j)})} \quad \text{(For simplicity, the constant is ignored.)}$$

$$= w(t-2\tau)^{(j)} \frac{p(\theta(t-\tau)^{(j)} | \mathbf{x}_{t-\tau})}{p(\theta(t)^{(j)})} \quad \text{(This is due to stationarity of } p(\theta(t)).) \tag{15}$$

Combine Eq. (14)-(15)and apply log to both side, we have:

$$\log \frac{w(t)^{(j)}}{w(t-\tau)^{(j)}} - \log \frac{w(t-\tau)^{(j)}}{w(t-2\tau)^{(j)}} = \log p(\theta(t)^{(j)} | \theta(t-\tau)^{(j)}) + \log(B)$$

Let $h(t) = \log \frac{w(t)^{(j)}}{w(t-\tau)^{(j)}}$. Due to the Markovian assumption, we have:

$$h(t) - h(t-\tau) = \int_{t-\tau}^{t} \left( \log p(\theta(s)^{(j)} | \theta(t-\tau)^{(j)}) \right)' ds \tag{16}$$

Given that $\tau$ is arbitrarily chosen and $\tau \in [0, \epsilon]$. Then for any $t \in [0, +\infty)$ we have:

$$h'(t) = \left( \log p(\theta(t)^{(j)} | \theta(t-\tau)^{(j)}) \right)' \tag{17}$$

Taking $\log \frac{w(t)^{(j)}}{w(t-\tau)^{(j)}}$ back to $h(t)$ and following Eq. (16)-(17), we have:

$$\left( \log w(t)^{(j)} \right)'' = \left( \log p(\theta(t)^{(j)} | \theta(t-\tau)^{(j)}) \right)''$$

$\square$

# B   Proof of Theorem 2

*Proof.* Let $t = \alpha \sqrt{-\log(-\beta)}$, where $\beta \in [-1, 0)$, we have:

$$\int_0^{+\infty} \pi(t) dt = \lim_{l \to 0} \int_{-1}^{l} \underbrace{\frac{-\alpha}{2\beta\sqrt{-\log(-\beta)}} \pi(\alpha\sqrt{-\log(-\beta)})}_{h(\beta)} d\beta$$

Let $h(\beta)$ be the integrand of the right-hand-side of the above equation, such that we can construct an $H : [-1, 0) \to [0, +\infty)$ that satisfies an initial value problem (IVP):

$$H'(\beta) = h(\beta), \quad H(-1) = 0$$

Then we have:

$$\int_0^{+\infty} \pi(t)dt = \lim_{l \to 0} H(l) = \lim_{l \to 0} \int_{-1}^l h(\beta)d\beta$$

Since $h(\beta)$ is not analytic at $\beta = 0$, we separate the solution into two parts:

$$\int_0^{+\infty} \pi(t)dt = \int_{-1}^{-\epsilon} h(\beta)d\beta + \lim_{l \to 0} \int_{-\epsilon}^l h(\beta)d\beta$$

$$= H(-\epsilon) + \lim_{l \to 0} \int_{-\epsilon}^l h(\beta)d\beta \tag{18}$$

We then construct two auxiliary IVPs to meet assumptions of Lemma 1 in [45]. Let $H_1, H_2 : [-\epsilon, 0) \to [0, +\infty)$ satisfy the initial value problem:

$$H_1' = h(\beta), \;\; H_1(-\epsilon) = H(-\epsilon)$$
$$H_2' = h(\beta - \epsilon), \;\; H_2(-\epsilon) = H(-2\epsilon)$$

By Lemma 1 [45], as $\epsilon \to 0$ we have:

$$\lim_{\epsilon \to 0} \left( \lim_{l \to 0} \int_{-\epsilon}^l h(\beta)d\beta \right) \leqslant \lim_{\epsilon \to 0} \|H(-\epsilon) - H(-2\epsilon)\| \tag{19}$$

Combining Eq.(18) and Eq.(19), we have:

$$\lim_{\epsilon \to 0} \left( \int_0^{+\infty} \pi(t)dt \right) = \lim_{\epsilon \to 0} \left( H(-\epsilon) + \lim_{l \to 0} \int_{-\epsilon}^l h(\beta)d\beta \right) \tag{20}$$

$$\leqslant \lim_{\epsilon \to 0} \left( H(-\epsilon) + \|H(-\epsilon) - H(-2\epsilon)\| \right) \tag{21}$$

Then we have:

$$\int_0^{+\infty} \pi(t)dt \leqslant \lim_{\epsilon \to 0} \left( H(-\epsilon) + \|H(-\epsilon) - H(-2\epsilon)\| \right) \tag{22}$$

$\square$

## C  Detailed Form of Training Loss

Our final loss (Eq. (12)) contains an ELBO term $l_b$:

$$l_b = \sum_{t=1}^{I \times N} (-\mathrm{KL}(\widetilde{q}(\theta(t)) \| p(\theta(t)) - \sum_{j=1}^k w(t)^{(j)} l_u(\cdot; \theta(t)^{(j)})), \tag{23}$$

where the KL term is calculated as:

$$\mathrm{KL}(\widetilde{q}(\theta(t)) \| p(\theta(t)) = \frac{\log(s_0(t)) - \log(s(t))}{2} + \frac{s(t) + (m(t) - m_0(t))^2}{2s_0(t)}. \tag{24}$$

We then follow ADDA [28] to calculate $l_u$ involved in the second term. Furthermore, $f(\cdot)$ in $l_{de}$ can be computed in closed-form:

$$f(m(t), m'(t), m''(t), s(t), s'(t), s''(t), \theta(t)^{(j)}) = \frac{1}{2(s(t))^3} [(s'(t))^2 s(t) - 2(m'(t))^2 (s(t))^2$$
$$+ (2m''(t)s(t) - 4s'(t)m'(t))(\theta_t^{(j)} - m(t))s(t)$$
$$+ (\theta_t^{(j)} - m(t))^2 (s''(t)s(t) - 2(s'(t))^2) - s''(t)s(t)].$$

## D  Background: Particle Filtering

Particle filtering (PF) provides an effective solution for state estimation problem of dynamical systems, in which sequential importance sampling (SIS) is used to approximate the evolution of the posterior distribution of the state given observations, $p(\theta_t | x_{1:t})$, where $\theta_t$ represents the latent state at time $t$

and $x_{1:t}$ denotes observations from time 1 to $t$. By assuming a Markov chain for the state space, PF adopts a weighted set of particles $\{(w_t^{(j)}, \theta_t^{(j)})\}_{j=1}^k$ to construct an approximate posterior:

$$q(\theta_t|x_{1:t}) = \sum\nolimits_{j=1}^{k} w_t^{(j)} \delta(\theta_t - \theta_t^{(j)}), \; \theta_t^{(j)} \sim q(\theta_t^{(j)}|\theta_{t-1}^{(j)}, x_{1:t}), \tag{25}$$

where $\delta$ denotes the Dirac-delta function; $\theta_t^{(j)}$ is the $j$-th particle (sample) drawn from an approximate distribution called importance distribution $q(\theta_t^{(j)}|\theta_{t-1}^{(j)}, x_{1:t})$, which can be assumed to have a simple parametric form, e.g. a Gaussian; $w_t^{(j)}$ is the corresponding importance weight, which can be calculated as:

$$w_t^{(j)} \propto w_{t-1}^{(j)} \frac{p(x_t|\theta_t^{(j)})p(\theta_t^{(j)}|\theta_{t-1}^{(j)})}{q(\theta_t^{(j)}|\theta_{t-1}^{(j)}, x_{1:t})}, \tag{26}$$

where $p(x_t|\theta_t^{(j)})$ is the likelihood.

By using transition distribution $p(\theta_t|\theta_{t-1})$ as the importance distribution, one can construct a bootstrap filter [17] with simplified importance weights as:

$$w_t^{(j)} \propto w_{t-1}^{(j)} p(x_t|\theta_t^{(j)}), \tag{27}$$

where $p(x_t|\theta_t^{(j)})$ is the likelihood and $\sum_j w_t^{(j)} = 1$. Practically, recursive multiplication of the previous weight usually leads to particle degeneracy, i.e., most particles having near-zero weights. To tackle this, sequential importance resampling [17] could be a better alternative. It is also worth noting that our method alleviates this problem by adopting neural networks to approximate the solution of importance weights in PFDE without the need to perform the recursive equation.

## E    Related Work: Particle Filtering with Neural Networks

Particle filtering (PF) [17] is a popular method for solving state estimation problems in time-varying systems. In essence, such a method adopts a general non-parametric form to represent the approximate posterior distribution of the latent state, whose distribution can be non-Gaussian and has a highly complex form. Due to its generalizability, PF has been applied in various research fields including robotics [46, 47, 48], signal processing [49, 50] and neuroscience [51].

Traditional PF usually adopts a large number of weighted particles to approximate the posterior distribution, which is computationally expensive. [52] manage to reduce the number of particles and propose an efficient PF algorithm by incorporating shallow neural networks. [53] further adopt PF to approximate the distribution of the hidden states of a deep recurrent network. Along a different line of research, Particle Flow [54] is proposed to transport particles from prior to posterior distribution using ordinary differential equations (ODEs). [55] augment particle flow with more powerful neural ODEs [35]. Though particle flows offer an effective solution for Bayesian filtering, their ODEs are defined over virtual time interval $[0, 1]$ and still assume discrete-time Markovian assumption for the state space. Thus, the posterior of the state can only be evaluated step by step along the Markov chain, requiring calculating the likelihood for observation at each time step. In contrast, ECBNN formulates PF with differential equations such that the posterior can be evaluated in real-time without requiring calculating the likelihood. Such property allows ECBNN to extrapolate model distributions over a time axis when only previous observations are available.

## F    Computational cost of ECBNN

Suppose ECBNN adopts K particles to represent the posterior of a neural network with M parameters. The computation complexity for ECBNN making predictions for N data samples is O(K*M*N), while the computation complexity for other offline UDAs, e.g. ADDA, is O(M*N). Notably, we adopt surrogate neural networks to approximate the solution of the differential equation, which is computationally efficient without the need for numerical integration. Therefore, the computation cost of solving the PFDE depends on the time cost of a single forward pass of the associated surrogate neural networks.

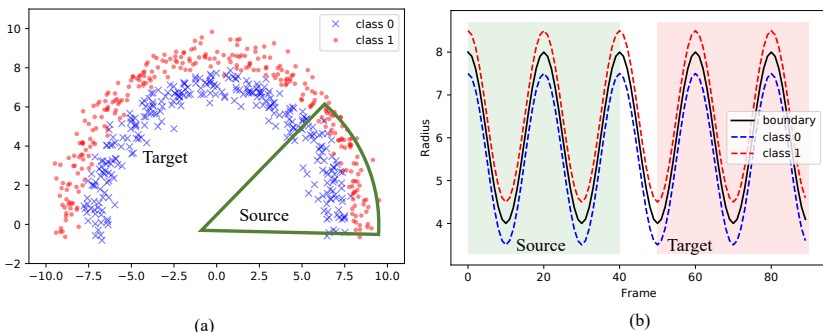

(a)                                         (b)

Figure 6: (a) Initial data samples at 0-th frame, where red dots and blue crosses are data samples from class 0 and class 1 respectively; Green sector indicates data samples from the source domain. (b) Visualization of motion dynamics of decision boundary and data, where Black solid line represents the decision boundary, the red hash line represents data for class 0, and blue hash line for class 1.

## G  Experiments

In this section, we provide more details of experiments and implementations of our ECBNN for each task. To formulate the streaming domain adaptation scenes, we split each sequence into equally-length batches, and each has ten frames. All experiments are performed on a machine running the Ubuntu operating system with an AMD EPYC 7302P 16-core CPU and an RTX A5000 GPU.

### G.1  Toy Dataset

#### G.1.1  Detailed Data Generation

We consider generating dynamic data points that have underlying time-varying distributions; this is facilitated by the polar coordinate system $(r, \theta)$. Specifically, we simulate time-varying distributions of the data samples by varying $r$ over time. For instance, the trajectory of the data sequence is represented by $(r(t), \theta)$. The initial data samples at the 0-th frame, e.g. $t = 0$, are shown in Fig. 6 (a), in which red dots represent data samples from class 0, and blue crosses represent class 1; the motion of decision boundary and the data samples are controlled by special form of $r(t)$. For example, we construct the time-varying decision boundary by $r(t) = \text{Acos}(wt) + \text{B}$, where the amplitude $\text{A} = 2$; the frequency $w = 2\pi/10$; the vertical shift $\text{B} = 6$; we move the data samples from class 0 by $r(t) = \text{Acos}(wt) + \text{B} + \text{abs}(\eta)$ and class 1 by $r(t) = \text{Acos}(wt) + \text{B} - \text{abs}(\eta)$, where $\eta$ are Gaussian random noises with standard deviation 0.5. This forces the data trajectory from class 0 to be outside the decision boundary and that from class 1 to be inside the decision boundary, as shown in Fig. 6 (b). We also add 2D isotropic Gaussian noises whose standard deviation is 0.2 to the locations of each sample. We take the 0-th to the 39-th frame to construct source data and the 50-th to the 89-th to construct target data. We create 2100 sequences as the training set, 300 sequences as the validation set, and 600 sequences as the testing set. Note that we still adopt Cartesian coordinates, e.g. (x,y), to represent the data samples during training and testing. Absolute time stamps $t$, radius $r(t)$ as well as angel $\theta$ are assumed to be unknown in our problem setting.

#### G.1.2  Detailed Training Procedure of ECBNN

Our ECBNN is trained by minimizing the loss (Eq. (12) in the main paper) using the Adam optimizer with a learning rate in the range of $[1 \times 10^{-4}, 1 \times 10^{-3}]$ and a weight decay of $5 \times 10^{-4}$ for 100 epochs. We set the hyperparameter $\lambda_d$ as 2. We search our model over $\lambda_{de}$ ranging from 0.0001 to 0.1 and $\lambda_i$ ranging from 0.01 to 0.1. Following the training strategy of $\beta-$VAE [56], we reweight the importance of the KL term in $l_b$ and set it as 0.5. We sample 14 particles during the experiments. The validation set is adopted to select the best-trained model.

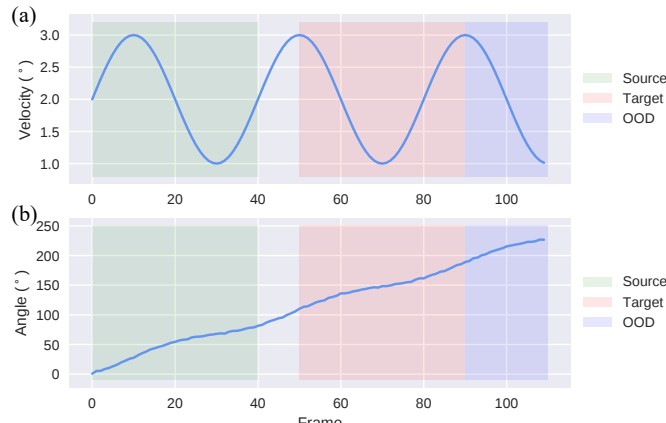

Figure 7: (a) Sinusoidal function to describe the angular velocity of streaming rotating MNIST and USPS (b) Trajectories of rotating angles

## G.2 Streaming Rotating MNIST→USPS

### G.2.1 Detailed Data Generation

We construct a synthetic video dataset based on MNIST and USPS: Streaming Rotating MNIST→USPS. The video sequences are generated from the rotating MNIST and USPS handwritten digits with sinusoidal angular velocity. The angular velocity is described by function $\sin(2\pi t/40) + 2$ (shown in 7). Consequently, as is shown in 7 (b), the rotating angle is monotonically increasing as time goes by. We also add Gaussian random noises whose standard deviation is $2/3°$ to each frame. We take the first 40 frames of the Streaming Rotationg MNIST video sequence as source data, whose rotating angles range from $0°$ to $80°$; we take 50-th to 90-th frames of the Streaming Rotationg USPS video sequence as target data, whose rotating angles range from $110°$ to $190°$; We also take 90-th to 110-th frames of the Streaming Rotationg USPS video sequence to construct an extra Out-of-Domain (OOD) testing set. In summary, we generate 5000 sequences for the training set, 1000 sequences for the validation set, 1000 sequences for the source-testing set, 1000 sequences for the target-testing set, and 1000 sequences for the OOD-testing set.

### G.2.2 Detailed Training Procedure of ECBNN

Since MNIST and USPS have different image sizes, we rescale each frame to $28 \times 28$ pixels. Our ECBNN is trained by minimizing the loss (Eq. (12) in the main paper) using the Adam optimizer with a learning rate in the range of $[1 \times 10^{-5}, 6 \times 10^{-5}]$ and a weight decay of $5 \times 10^{-4}$ for 50 epochs. The hyperparameters balancing among losses are set as $\lambda_d = 0.5$, $\lambda_{de} = 2 \times 10^{-4}$ and $\lambda_i = 1.0$. The importance of the KL term is reweighted to be $1.1$. Besides, the number of sampled particles is 12. The validation set is adopted to select the best-trained model.

### G.2.3 Detailed Implementation of ECBNN

(1) **Inference**: To infer the posterior distribution of NN parameters over time by approximating solution of PFDE, we adopt three neural networks: $\{f_{\theta_m}, f_{\theta_s}, f_{\theta_w}\}$. We adopt a similar achitecture of Encoder (which we discussed next) to process the historical batches $\mathbf{B}_{1:i-1}$, followed by three branches for calculation of $m(t)$, $s(t)$ and $w(t)^{(j)}$. The branches for $f_{\theta_m}$ and $f_{\theta_s}$ have similar architecture designs: we adopt three residual blocks except that $f_{\theta_s}$ needs a Softplux activation before the output; each residual block is composed of two fully connected layers and a residual connection. Then we sample the particles $\theta(t)^{(j)}$ using $\mathcal{N}(m(t), s(t))$. We then use particles to update the NN parameters of the encoder (which we discussed next). These particles are further adopted as extra inputs of the branch for the calculation of $w(t)^{(j)}$.

(2) **Encoder**: We adopt an extension of LeNet [57] to implement the encoder $f_e$. Firstly, taking as input the video frames, two 2D convolutional layers whose output channels are 20 and

50, respectively are adopted; the output dimension of the second convolutional layer is $50 \times 4 \times 4$; each convolution is followed by a max pooling layer and ReLU activation. After that, we adopt a fully connected layer with 16 hidden nodes to each channel, whose parameters are updated using particles drawn from the posterior from PFDE. We adopt another fully connected layer to reduce the dimension of feature maps into the dimension of $z$ (Fig 1. in the main paper). Finally, a 2-layer GRU module is utilized to capture the temporal dynamics. We apply a dropout rate of 0.5 in $f_e$ to avoid over-fitting.

(3) **Prior**: The prior $p(\theta(t))$ is computed through neural networks $\{f_{\theta_{m0}}$ and $f_{\theta_{s0}}\}$. Similarly, we adopt a similar architecture of Encoder of DA to process the current batches $\mathbf{B}_i$, followed by two branches for calculation of $m_0(t)$ and $s_0(t)$. Both branches contain a fully connected layer with 256 hidden neurons. The activation for the branches computing $s_0(t)$ is Softplus activation.

(4) **Predictor**: The predictor $f_p$ consists of a fully connected layer with 256 hidden nodes, followed by a batch normalization layer and ReLU activation, followed by a fully connected layer, whose output dimension equals to the number of classes.

(5) **Discriminator**: We follow ADDA [28] to implement discriminator $f_d$, whose architecture contains three fully connected layers. The first two layers have the hidden nodes twice as the dimension of $z$ and are followed by BN and ReLU activations. The output of the third layer is 1-dimensional, followed by a Sigmoid activation.

### G.3 Multi-view Lip Reading

### G.3.1 Detailed Training Procedure of ECBNN

Our ECBNN on the OuluVS2 dataset is trained using the Adam optimizer with a learning rate of $2 \times 10^{-4}$ for 100 epochs. For the hyperparameters balancing among the loss terms: $\lambda_d$ is set as 1.0; both $\lambda_{de}$ and $\lambda_i$ are set as 0.05. The importance of the KL term is reweighted to be 1.1, and the number of sampled particles is 12.

### G.3.2 Detailed Implementation of ECBNN

(1) **Inference and Prior**: We adopt the same architecture design of $\{f_{\theta_m}, f_{\theta_s}, f_{\theta_w}\}$ and $\{f_{\theta_{m0}}, f_{\theta_{s0}}\}$ as stated in section Sec. G.2.3 (Detailed Implementation of ECBNN).

(2) **Encoder**: We follow the neural network architecture of [44] in our implementation of the encoder $f_e$. Firstly, three 3D convolutional layers whose output channels are 32, 64, and 96, respectively, are adopted as the backbone. The output dimension is $96 \times 1 \times 3 \times 3$. Then we adopt a fully connected layer with nine hidden nodes to each channel, whose parameters are updated using particles drawn from the posterior from PFDE. Then we flatten the tensor and feed it into three fully connected hidden layers of sizes 2000, 1000, and 500, respectively, followed by a linear bottleneck layer. The $\Delta$ (first derivatives) and $\Delta\Delta$ (second derivatives) features are appended to the bottleneck layer, followed by an LSTM with 450 hidden states. A dropout rate of 0.5 is used.

(3) **Predictor**: The predictor $f_p$ is implemented by a fully connected layer. Its output size equals to the number of classes, in this case, 10. We then adopt Softmax to transform logits into the probabilities over all classes.

(4) **Discriminator**: The discriminator $f_d$ is composed of three $1 \times 1$ convolutional layers whose output channels are all 512. Each convolution is followed by batch normalization and a LeakyReLU activation. Then we flatten the tensor and feed it to a fully connected layer with an output size of 1. The final output is activated by Sigmoid.

### G.3.3 Detailed results on OuluVS2 test set

We repeat the training procedure across 3 different random seeds and report the detailed accuracy on the OuluVS2 testing set with respect to 4 view angels in Table 4. Again, we can see that ECBNN consistently outperforms other baselines for all views.

Table 4: Detailed Accuracy (%)(mean± std) on the OuluVS2 test set at source and each target domain

| Method | 30° | 45° | 60° | 90° | Average |
|---|---|---|---|---|---|
| Source-Only | $73.4 \pm 2.2$ | $60.1 \pm 1.1$ | $41.1 \pm 1.3$ | $8.4 \pm 4.3$ | $45.8 \pm 0.2$ |
| DANN [27] | $75.5 \pm 0.5$ | $74.2 \pm 1.6$ | $67.6 \pm 0.5$ | $65.2 \pm 2.8$ | $70.6 \pm 0.7$ |
| ADDA [28] | $78.3 \pm 1.8$ | $71.9 \pm 1.8$ | $68.2 \pm 0.1$ | $58.7 \pm 3.3$ | $69.3 \pm 0.5$ |
| CIDA [30] | $80.2 \pm 2.3$ | $77.0 \pm 2.2$ | $68.8 \pm 2.7$ | $23.0 \pm 5.3$ | $62.3 \pm 0.9$ |
| DSAN [29] | $77.3 \pm 0.5$ | $76.4 \pm 1.6$ | $72.2 \pm 1.6$ | $37.4 \pm 6.0$ | $65.8 \pm 1.6$ |
| EDA [11] | $78.4 \pm 1.5$ | $72.5 \pm 0.8$ | $67.7 \pm 1.3$ | $62.9 \pm 0.6$ | $70.3 \pm 0.2$ |
| ECBNN (Ours) | $\mathbf{81.8 \pm 0.5}$ | $\mathbf{76.1 \pm 2.4}$ | $\mathbf{71.1 \pm 3.0}$ | $\mathbf{67.5 \pm 1.1}$ | $\mathbf{74.1 \pm 0.9}$ |

### G.3.4 Training Time

We report the training time per epoch on OuluVS2 for our ECBNN and baselines. We conduct the timing experiments using the PyTorch package. All experiments are performed on a Ubuntu machine with an AMD EPYC 7302P 16-core CPU and an RTX A5000 GPU. Each model took around 50 epochs, and their average running time is reported. As shown in Table 5, the offline UDA baselines, including DANN, ADDA, CIDA, and DSAN, run almost two times faster than the online UDA baseline, EDA. The training time of ECBNN is in the same magnitude as that of EDA, though the latter runs three times faster than the former. It is also worth noting that for a fair comparison, all evaluated models have the same number of parameters.

Table 5: Training time (s) of our ECBNN and baselines on OuluVS2 dataset

| Method | Training time (s) ↓ |
|---|---|
| Source-Only | 12.71 |
| DANN [27] | 14.55 |
| ADDA [28] | 14.54 |
| CIDA [30] | 14.42 |
| DSAN [29] | 16.32 |
| EDA [11] | 23.81 |
| ECBNN (Ours) | 70.30 |

## H   More Visualization Results on Toy Dataset

We visualize the predicted classification boundaries generated by different methods at two more frames, shown in Fig 8.

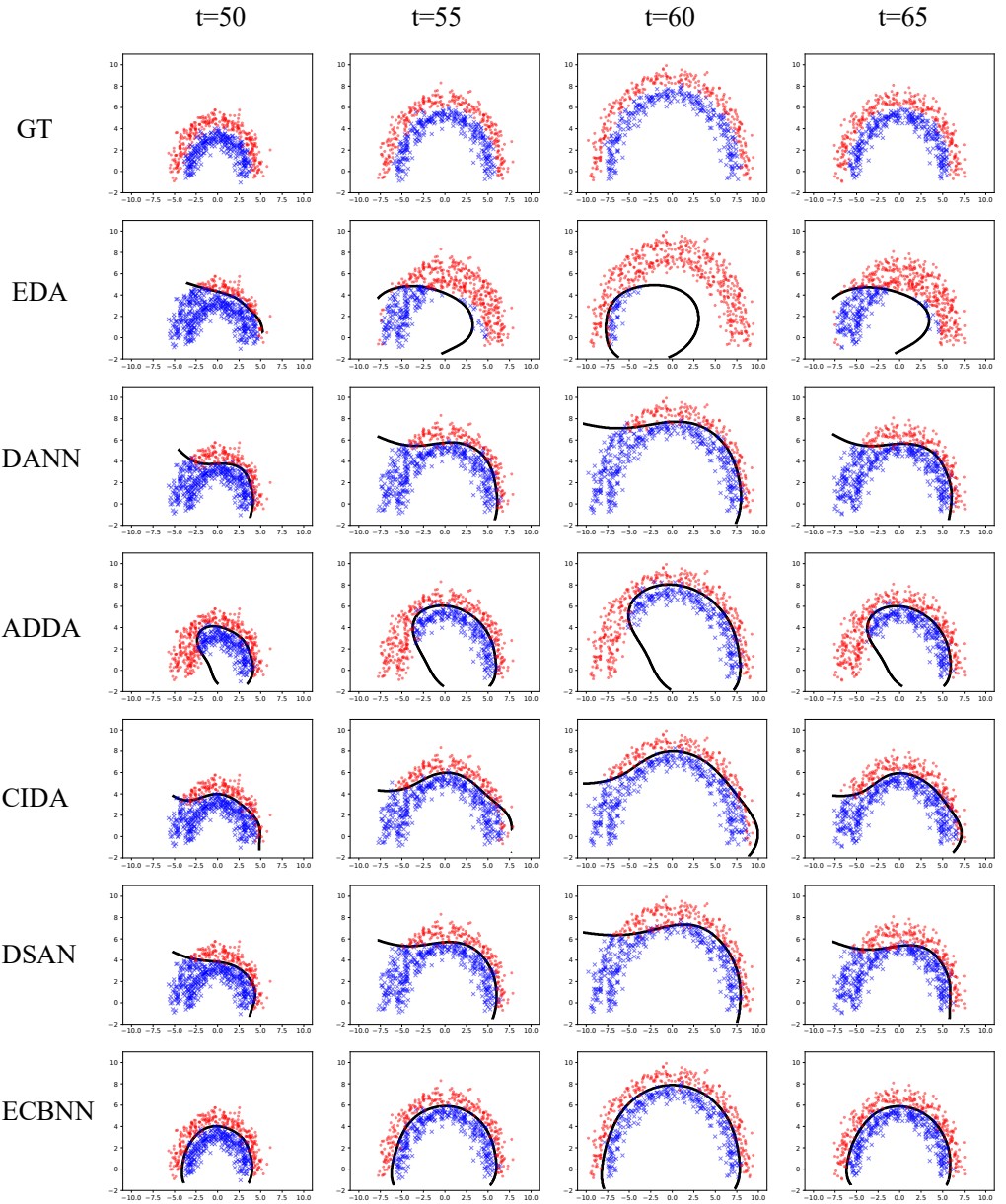

Figure 8: Results on the Growing Circle dataset. Black lines represent the decision boundaries generated according to model predictions.