# OpenReview forum: "Extrapolative Continuous-time Bayesian Neural Network for Fast Training-free Test-time Adaptation"
_NeurIPS.cc/2022/Conference — NeurIPS 2022 Accept_

### Official Review · Reviewer_dicg · 2022-07-09

**Rating:** 6
**Confidence:** 3
**Soundness:** 4 excellent
**Presentation:** 3 good
**Contribution:** 3 good

**Summary:**

The paper describes a low-latency unsupervised domain adaptation (UDA) framework for non-stationary streaming data. To this end, the method includes a Bayesian neural network, which produces a temporally evolving parameterization to the feature extractor, acting as a meta network with temporally changing output. It includes furthermore a discriminator, allowing adversarial domain adaptation training. Thirdly a predictive module is fed by the invariant representation. The Bayesian inference is tackled via Variational Inference relying on a continuous time particle filtering idea.


**Questions:**

Please clarify what is happening during training. Figure 1 suggests that two embedding is generated $\tilde{z}$ and $z$. Two interpretation is possible, either $z$ corresponding to the source domain, and $\tilde{z}$  to the target domain. Therefore both go to the discriminator and only $z$ goes to the predictor.
Other interpretation would be that for both domain, two embeddings are generated, resulting in $z^s$, $z^a$, $\tilde{z}^s$ and $\tilde{z}^a$. The statement in Line 135-136 calling $\tilde{z}$ as "the encoder output for the approximated data" not helps this confusion. What is "approximate data"?

Is $l_u(\cdot ; \theta)$ itself an extreme value (as described by Equation 4), which later enter to the loss, describing a multilevel optimization problem? Or $l_u(\cdot ; \theta)$  supposed to be the function inside the saddle point optimization? (I assume the second is true. If so please fix the equation)

Equation 5 have two more opening brackets than closing brackets, it is possible to guess what this equation was intended to be, but please fix this. (Eq. 6 also missing a closing bracket)

Please give a more detailed motivation for using $l_i$ temporal-domain-invariance loss besides the $V_d$ term in $l_u$. Even better would be to execute an ablation study.







**Limitations:**

The authors describe that their method is only capable to work with continuously evolving systems at the moment, and they mention handling discrete dynamics as a future work.

It is not entirely clear how large datasets can be handled. The method is quite complex. The evaluation time computational complexity seems quite favorable (Table 2). The training time complexity can be interesting as well.

**Strengths And Weaknesses:**

Domain adaption  on non-stationary stream in real time is unquestionably a problem with both theoretical and practical significance.
The described framework clearly novel and the description of the method is clear.  Despite the clarity of the description the paper is quite dense, as the method is quite complex. It takes 5 or 6 equations to describe its loss function alone. This level of complexity requires further justification. Ablation studies are entirely missing from the paper. The question inevitably appears in the reader, does this complexity really necessary?

It is not entirely clear why Bayesian approach is necessary. It is easy to imagine that it helps, it is just far from being trivial that it does. What are the weak points of an evolving point estimate approach versus an evolving Bayesian approach for estimating the encoder parameters?

The problem setup could be described with a bit more clarity, as it is not always entirely clear what is the main difference between the source and target domain. In the growing circles dataset for example, Fig 2A shows that the source corresponds to a green wedge in the input space, and the target is the other part. Fig 2B shows, that the Source and the Target is temporally separated (0 -39th frame and 50-89th frames respectively). Is it both?

L210-212 describes an unusual solution, the prior is used more like a target, where the KL term in the variational lower bound is used at training time to force the estimated parameters to follow this target. This use seems against the meaning of prior in Bayesian framework.

---

> ### Author Response · Authors · 2022-08-02
> **[3/3] Thanks and response to concerns**
>
>
> **References:**
>
> [a] Chung, J., Kastner, K., Dinh, L., Goel, K., Courville, A. C., & Bengio, Y. (2015). A recurrent latent variable model for sequential data. Advances in neural information processing systems, 28.
>
> [b] He, H., Wang, H., Lee, G. H., & Tian, Y. (2018, September). Probgan: Towards probabilistic gan with theoretical guarantees. In International Conference on Learning Representations.
>
> [c] Izmailov, P., Vikram, S., Hoffman, M. D., & Wilson, A. G. G. (2021, July). What are Bayesian neural network posteriors really like?. In International conference on machine learning (pp. 4629-4640). PMLR.

---

> ### Author Response · Authors · 2022-08-02
> **[2/3] Thanks and response to concerns**
>
>
> **Q2: “It is not entirely clear why Bayesian approach is necessary…. What are the weak points of an evolving point estimate approach versus an evolving Bayesian approach for estimating the encoder parameters?”**
>
> **A2:** Inspired by internal predictive modeling in neuroscience, we aim to model time-evolving neural network parameters of UDA given non-stationary streaming data. Such a problem cannot be suitably handled by a point estimate approach:
>
> (i) Point estimate approaches usually adopt maximum likelihood estimation (MLE) for training, requiring labels for specific tasks, e.g. classification, and regression.
>
> (ii) Real-world streaming data are usually characterized by complex temporal dynamics. However, the annotations of such dynamics are usually unavailable.
>
> (iii) We are only given the streaming data, but time-evolving neural network parameters are unknown. Again, it is difficult for a point estimate approach to directly learn such time-evolving parameters from the streaming data.
>
> In contrast, our evolving Bayesian framework addresses this problem by inferring a stochastic dynamical systems given the streaming data, allowing us to reason the unknown dynamics of the streaming data and neural network parameters.
>
> - - -
>
> **Q3: “ …it is not always entirely clear what is the main difference between the source and target domain…”**
>
> **A3:** We are sorry for the confusion. The main differences between the source and target domain are:
>
> + They have distinct categorical domains.
>
> + They may not share the same timeline.
>
> For instance, in the experiment on the Growing Circle dataset, the source and target domains are *both spatially and temporally separated* (as shown in Figure 2A and 2B).
> - - -
>
> **Q4: “ …the prior is used more like a target…This use seems against the meaning of prior in Bayesian framework.”**
>
> **A4:** Recall that we force the inference of variables involved in PFDE (Eq. (7)) to depend on **historical data batches** only. To encourage the posterior (produced by using these variables) of NN parameters to **extrapolate the future**, we take a further step and adopt the **current** batch to estimate the **prior for “future parameters”**. It’s worth noting that using data and neural networks to estimate the parameter prior is a common practice in Bayesian deep learning methods [a,b,c].
> - - -
>
> **Q5: “Please clarify what is happening during training… Other interpretation would be that for both domain, two embeddings are generated…What is "approximate data"?”**
>
>  **A5:** We are sorry for the confusion. $z$ and $\tilde{z}$ are generated for both source and target domains; that is your second interpretation is correct. By approximated data, we are referring to $\tilde{ {\bf x}_t }(\phi)$ in Eq. (10); it is used to calculate the upper bound of the temporal-domain-invariant loss. Thanks to Theorem 2, such data can be approximated by simply using linear interpolation over the hidden representation of the current batch.
> - - -
> **Q6: “ The evaluation time computational complexity seems quite favorable (Table 2). The training time complexity can be interesting as well.”**
>
> **A6:** Thanks for the constructive suggestion.
> Table 2 below shows the training time per epoch on OuluVS2 for our ECBNN and baselines. We conduct the timing experiments using the PyTorch package. All experiments are performed on an Ubuntu machine with an AMD EPYC 7302P 16-core CPU and an RTX A5000 GPU. Each model took around 50 epochs, and their average running time is reported.
>
> Table2: Training time (in seconds) of our ECBNN and baselines on OuluVS2 dataset.
> | Method | Training time per epoch(s)|
> | :------------| :-----------:|
> | Source-Only | 12.71 |
> | DANN [15] | 14.55 |
> | ADDA [16] | 14.54 |
> | CIDA [18] | 14.42 |
> | DSAN [17] | 16.32 |
> | EDA [11] | 23.81 |
> | ECBNN (ours) | 70.3 |
>
> As shown in Table 2, the offline UDA baselines, including DANN, ADDA, CIDA and DSAN, run almost 2 times faster than the online UDA baseline, EDA. The *training time* of ECBNN is in the same magnitude as that of EDA, though the latter runs 3 times faster than former. Although our ECBNN takes more time to train, it enjoys much shorter evaluation/inference time, i.e., 0.01s for ECBNN versus 0.23s for EDA. Therefore as you mentioned, “the evaluation time computational complexity seems quite favorable (Table 2)”. It is also worth noting that for fair comparison, all evaluated models have the same number of parameters.
> - - -
>
> **Q7: “ Or  $l_{u}(\cdot;\theta)$  supposed to be the function inside the saddle point optimization? ”**
>
> **A7:** Sorry for the confusion.  Indeed, $l_{u}(\cdot;\theta)$ should be the function inside the saddle point optimization. We will make this clear in the revision.

---

> ### Author Response · Authors · 2022-08-02
> **[1/3] Thanks and response to concerns**
>
> Thank you for the careful review and constructive suggestions.
>
> We answer specific questions below:
>
> **Q1.1: “ …It takes 5 or 6 equations...This level of complexity requires further justification…”**
>
> **A1.1:** This is a good suggestion. We justify the complexity as follows.
>
> Due to the drifting nature of the non-stationary streaming data, aligning features extracted across source and target streams is critical yet challenging for the success of unsupervised domain adaptation (UDA). Simply aligning using partial observation of the time series or the local streams (which can be achieved by $V_d$ in the vanilla UDA) leads to poor alignment quality. To address this challenge, we proposed to perform UDA on *the entire data generation mechanism*, which is described by a latent stochastic process conditioned on historical data. The corresponding loss is the  temporal-domain-invariance loss, $l_i$ (Eq.(9)).
>
> This leads to another challenge: aligning two latent stochastic processes tends to result in an intractable discriminator loss. To address the second challenge, we further derive an analytical upper bound for the discriminator loss, namely, the upper bound of temporal-domain-invariance loss, $\tilde{l}\_i$ (Eq.(11)).
>
> However, calculating such a bound requires evaluating posteriors of the UDA encoder in continuous time; this then leads us to formulate the problem using particle filter differential equation (PFDE), rather than traditional Bayesian filtering methods, such as particle filters(Reviewer jpUT also agreed on this and acknowledged that PFDE is ``"an original contribution"`` ``"with strong motivation for efficient multi-step ahead inference"``). We will include the discussion above in the revised version as suggested.
> - - -
>  **Q1.2: “ …Ablation studies are entirely missing from the paper…”,  “Please give a more detailed motivation for using $l_i$, temporal-domain-invariance loss besides the $V_d$ term in $l_u$. Even better would be to execute an ablation study.”**
>
> **A1.2:**
>
> *Motivation for Using $l_i$*
>
> Due to the drifting nature of the non-stationary streaming data, aligning features extracted across source and target streams is critical yet challenging for the success of unsupervised domain adaptation (UDA). Simply aligning using partial observation of the time series or the local streams (which can be achieved by $V_d$ in the vanilla UDA) leads to poor alignment quality. To address this challenge, we proposed to perform UDA on *the entire data generation mechanism*, which is described by a latent stochastic process conditioned on historical data. The corresponding loss is the  temporal-domain-invariance loss, $l_i$ (Eq.(9)).
>
> *Ablation Study*
>
> We think it is a good suggestion to perform an ablation study for verifying the above statement. Our final loss has three parts, namely the ELBO, $l_b$, PFDE loss, $l\_{de}$, and the upper bound of the temporal-domain-invariant loss, $\tilde{l}\_i$. Therefore, according to your suggestion, we then added an ablation study on Streaming Rotating MNIST $\rightarrow$ USPS dataset to verify the effectiveness of each part.
>
> Table1: Ablation study results ( accuracy (\%) ) on Streaming Rotating MNIST $\rightarrow$ USPS dataset
>
> | Method | Target Domain| Out-of-Domain|
> | :------------| :-----------:| :-:|
> | ECBNN (ours) | 60.9 |38.5|
> | w/o $\tilde{l}\_i$ | 53.6 | 29.9 |
> | w/o $\tilde{l}\_i$ & w/o $l\_{de}$ | 43.2 | 28.0 |
>
> Table 1 shows the frame-level classification accuracy on the target-testing set and the out-of-domain-testing set (OOD-testing set), from the target domain and “out-of-domain”, respectively. We can see that by removing the upper bound of the temporal-domain-invariant loss, $\tilde{l}\_i$, such that only $V_d$ term is contained in the discriminator loss,  the accuracy drops dramatically for both the target domain and out-of-domain. The accuracy drop is more severe for “out-of-domain”, verifying the effectiveness of our temporal-domain-invariant loss, $\tilde{l}\_i$, in generalizing to unseen streaming data. Moreover, the accuracy drops further for both target domain and out-of-domain after removing the PFDE loss, $l\_{de}$; this verifies the effectiveness of the PFDE loss.

---

> ### Comment · Reviewer_dicg · 2022-08-03
> **I need some clarification related to Question 2**
>
> Thank you for your detailed answers. For most of the questions they were clear to me, and clarified a few point. I have still a question about Q2. You claim Bayesian treatment was used because
>
> (i) Point estimate approaches usually adopt maximum likelihood estimation (MLE) for training, requiring labels for specific tasks, e.g. classification, and regression.
>
> There is nothing in the Bayesian and point estimate distinction what would prescribe the requirement of labels or using of MLE criterion. Every Bayesian learning problem defines a posterior, therefor a maximum a posterior estimate (MAP) always exist. If you can formulate your problem without direct supervision in a Bayesian way, you always can do it in MAP way.
>
> (iii) I do not see the problem of predict the parameters using a non Bayesian meta network, and backpropagate through the dynamically parameterized network, so no extra labels are needed for the meta network.
> Am I missing something here?
>
> Again I do not question the benefits of Bayesian methods in general, i am a practitioner of them myself. I would like a bit of motivation here, and the given motivation cited above (point estimation would need labels) I am afraid incorrect.
> I am happy to be corrected however.

---

> > ### Author Response · Authors · 2022-08-03
> > **Thanks for your follow-up question and for keeping the communication channel open.**
> >
> > Thanks for your follow-up question and for keeping the communication channel open. We noticed our mistake and agreed that a non-Bayesian meta network could also be used to dynamically predict neural network parameters. We want to restate our motivation as follows:
> >
> > + Real-world non-stationary streaming data usually are highly uncertain in their temporal dynamics. In this case, point estimate approaches could have difficulty encoding uncertainty of the temporal dynamics due to their deterministic nature; therefore, our evolving Bayesian approach could be a better option.
> >
> > + The predicted future neural network parameters are possibly multimodal given the historical non-stationary streaming data. For instance, consider recognizing words from lip movements when the human head is moving, and view angels are dynamically changing. Due to the uncertainty of the head movements, there could be multiple possible view angels at the next step, given the historical observations. Due to such “multiple possible futures”, ideally the predicted future distribution of neural network parameters should be multimodal as well. However, this may pose a significant challenge for point estimate approaches to encoding such multimodal distribution. In contrast, our evolving Bayesian framework adopts a weighted set of particles to represent the posterior distribution (Eq. (3)), which can represent distributions with arbitrary shapes using a sufficient number of particles.

---

### Official Review · Reviewer_jpUT · 2022-07-11

**Rating:** 7
**Confidence:** 2
**Soundness:** 4 excellent
**Presentation:** 3 good
**Contribution:** 4 excellent

**Summary:**

The paper proposes an approach to unsupervised domain adaptation by formulate predictive modeling as a continuous-time Bayesian filtering problem. The works introduces extrapolative continuous-time bayesian neural networks with a particle filtering differential equation at its core that only rely on historical data for inference and models the change of importance weights. The proposed model is trained in a framework that combines the minmax optimization of unsupervised domain adaptation and the ELBO of VAE. To  apply the model to non-stationary streaming data, an additional temporal domain invariant loss term was used to encourage the model to generalize to unseen data. Experiment results show strong performance of the proposed ECBNN against other unsupervised domain adaptation models.

**Questions:**

An important component of many discrete-time particle filters is to resample the particles to remove particles of low weights based on effective sample size. Will the proposed filtering approach based on PFDE suffer from the similar problem where the weights are concentrated on a few particles?

**Limitations:**

The author address the limitations of the work but not discusses negative social impact. But I think it might not be required for this work which is a primarily methodology paper.

**Strengths And Weaknesses:**

Strength:
Treating the time-evolving NN parameters as a continuous-time stochastic dynamic system and extending bayesian neural networks to continuous-time domains is a novel approach to unsupervised domain adaptation. I also appreciate the proposal of PFDE. It is an original contribution to the reviewer’s best knowledge with strong motivation for efficient multi-step ahead inference. The experiment settings considered by the paper are comprehensive and diverse

Weakness:
A minor concern I have about the paper is the organization of the method section. The traditional particle filtering method introduced in the background section is not part of the original contribution of the work and may be better incorporated in the related works section.

---

> ### Author Response · Authors · 2022-08-02
> **Thanks and response to concerns**
>
> Thank you for your encouraging and constructive comments. For better readability for other reviewers, we have decided to keep the Section/Subsection structure unchanged currently. However, we promise to reorganize the paper by combining related works and background section in the revised version.
>
> For the posed question:
>
> **Q1: “ Will the proposed filtering approach based on PFDE suffer from the similar problem where the weights are concentrated on a few particles?”**
>
> **A1:** This is a good question. Indeed, particle degeneracy is an important issue for traditional particle filtering methods. Taking bootstrap filters as an example, the importance weights are recursively updated as: $ w_{t}^{(j)} \propto  w_{t-1}^{(j)} p(x_t |\theta_t)$. Due to the recursive nature of the update equation, $w_{t}^{(j)}$ tends to be near-zero if the previous importance weight $w_{t-1}^{(j)}$ is near-zero, consequently leading to the particle degeneracy problem when the sequence is sufficiently long. Our method alleviates this problem by adopting neural networks to approximate the solution of importance weights in PFDE, without the need to perform the recursive equation. We will add a similar discussion in the revised version.

---

### Official Review · Reviewer_ZNi1 · 2022-07-12

**Rating:** 5
**Confidence:** 3
**Soundness:** 3 good
**Presentation:** 2 fair
**Contribution:** 3 good

**Summary:**

This paper proposes an adaptive unsupervised domain adaptation based on a variational particle filter Bayesian posterior on an encoder, which is then used to form predictions on a non-stationary (time changing) task. The particle filter is run with respect to a continuous-time differential equation and some theoretical results are proposed to associate the particle weights with the parameter-space transition distribution, used to form one of the terms of the training objective. Some comparative experimental results are presented.

**Questions:**

- why do we need a differential equation to model the trajectory of theta? what is the advantage of this burdensome choice?
- clarify what '' means in Thm 1 before it is used (2nd derivative wrt time, not one page later)
- I do not understand the last line 1 of page 1 (supp. mat.) of the proof of Theorem 1: p(x_t) may be an invariant distribution function wrt t but it is not a constant wrt the *value* of x_t.
- how does the proposed algorithm scale up wrt things like the number of parameters of the encoder, the number of training examples, and other relevant quantities? How about the computational cost of solving the differential equation?
- in the context of line 120, I would like to say that offline adaptation is biologically plausible, since we know that consolidation from hippocampus to cortex is taking place during sleep and may be a form of offline adaptation; if this is good enough for animals, why not for AI tasks?
- sec. 3.3 did not make sense to me, needs more work to clarify
- eq 8: does it make sense to have the current batch B_i as an input to compute a Bayesian prior?
- l. 213 I did not understand why this should be true



**Limitations:**

- computational costs are not analyzed
- results do not compare against the family of neural processes (which have similar properties)


**Strengths And Weaknesses:**

Pros:
- this paper attempts to address many difficult questions, including an efficient prediction of the Bayesian parameter posterior, do it in a non-stationary environment (albeit assuming it is stationary which is weird), and doing it with neural networks and solving differential equations to capture the trajectory of the posterior
- comparative results (although not against all the relevant baselines) seem impressive

Cons:

- The overall system seems very complicated and I cannot say that I was able to get an overall satisfying grasp of it
- I have some concerns about the proof of Theorem 1.
- I have some concerns that the posterior is a unimodal Gaussian over parameters
- in terms of related work and baselines, it seems that this paper has a blind spot wrt the family of conditional neural processes architectures (which also propose an efficient way to directly generate samples from the parameter posterior, conditioned on the past training data).
- there is a bit of a contradiction between the UDA setup where new tasks can appear at any time and the assumption of stationarity of p(theta(t)) assumed for Thm 1.
- a much more transparent analysis of different computational costs is needed

---

> ### Author Response · Authors · 2022-08-02
> **[3/3] Thanks and response to concerns**
>
> **Q5: “does it make sense to have the current batch B_i as an input to compute a Bayesian prior?”**
>
> **A5:**  Recall that we force the inference of variables involved in PFDE (Eq. (7))  to depend on **historical data batches** only. To encourage the posterior (produced by using these variables) of neural network parameters to **extrapolate the future**, we take a further step and adopt the **current** batch to estimate the **prior for “future parameters”**. It’s worth noting that using data and neural networks to estimate the parameter prior is a common practice in Bayesian deep learning methods [f,g,h].
>
> - - -
>
> **Q6: “l. 213 I did not understand why this should be true”**
> I.213: It’s worth noting that this will not affect the latency as prior will be removed during testing.
>
> **A6:** We adopt the variational inference to optimize the ECBNN. The prior, which depends on the current batch, is only required for computing the ELBO during training. Specifically, the KL divergence between the prior and the variational distribution in the ELBO can be seen as a regularization term during training. Such a term is no longer needed during testing (inference). As a result, the neural networks for estimating the prior are removed during testing. Therefore our model no longer needs to wait for the arrival of the current batch, without affecting the latency.
>
>
> - - -
>
>
> **References:**
>
> [a] Garnelo, M., Rosenbaum, D., Maddison, C., Ramalho, T., Saxton, D., Shanahan, M., ... & Eslami, S. A. (2018, July). Conditional neural processes. In International Conference on Machine Learning (pp. 1704-1713). PMLR.
>
> [b] Gordon, J., Bruinsma, W. P., Foong, A. Y., Requeima, J., Dubois, Y., & Turner, R. E. (2019). Convolutional conditional neural processes. arXiv preprint arXiv:1910.13556.
>
> [c] Foong, A., Bruinsma, W., Gordon, J., Dubois, Y., Requeima, J., & Turner, R. (2020). Meta-learning stationary stochastic process prediction with convolutional neural processes. Advances in Neural Information Processing Systems, 33, 8284-8295.
>
> [d] Markou, S., Requeima, J., Bruinsma, W. P., Vaughan, A., & Turner, R. E. (2022). Practical Conditional Neural Processes Via Tractable Dependent Predictions. arXiv preprint arXiv:2203.08775.
>
> [e] Raissi, M., Perdikaris, P., & Karniadakis, G. E. (2019). Physics-informed neural networks: A deep learning framework for solving forward and inverse problems involving nonlinear partial differential equations. Journal of Computational physics, 378, 686-707.
>
> [f] Chung, J., Kastner, K., Dinh, L., Goel, K., Courville, A. C., & Bengio, Y. (2015). A recurrent latent variable model for sequential data. Advances in neural information processing systems, 28.
>
> [g] He, H., Wang, H., Lee, G. H., & Tian, Y. (2018, September). Probgan: Towards probabilistic gan with theoretical guarantees. In International Conference on Learning Representations.
>
> [h] Izmailov, P., Vikram, S., Hoffman, M. D., & Wilson, A. G. G. (2021, July). What are Bayesian neural network posteriors really like?. In International conference on machine learning (pp. 4629-4640). PMLR.

---

> > ### Comment · Reviewer_ZNi1 · 2022-08-05
> > **Thanks for rebuttal details**
> >
> > I want to thank the authors for taking my concerns seriously and responding to each of my questions. I have upgraded my ratings.

---

> ### Author Response · Authors · 2022-08-02
> **[2/3] Thanks and response to concerns**
>
> We answer specific question below:
>
> **Q1:  “I have some concerns that the posterior is a unimodal Gaussian over parameters”**
>
> **A1:**  Thank you for mentioning the posterior of neural network parameters. Actually, our posterior distribution is not a unimodal Gaussian. This is because our PFDE adopts a weighted set of particles to represent the posterior distribution (Eq. (3)), which can represent distributions with arbitrary shapes using a sufficient number of particles.
> - - -
> **Q2: “a much more transparent analysis of different computational costs is needed” , “how does the proposed algorithm scale up wrt things like the number of parameters of the encoder… How about the computational cost of solving the differential equation?”**
>
> **A2:** Thank you for the constructive suggestion. Suppose ECBNN adopts K particles to represent the posterior of a neural network with M parameters. The computation complexity for ECBNN making predictions for N data samples is O(K*M*N), while the computation complexity for other offline UDAs, e.g. ADDA, is O(M*N). Following [e,30] for solving the proposed PFDE, we adopt surrogate neural networks (Eq. (7)) to approximate the solution of the differential equation, which is computationally efficient without the need for numerical integration. Therefore, the computation cost of solving the PFDE depends on the time cost of a single forward pass of the associated surrogate neural networks.
> - - -
> **Q3: “ why do we need a differential equation to model the trajectory of theta? what is the advantage of this burdensome choice?”**
>
> **A3:** Due to the drifting nature of the non-stationary streaming data,  aligning features extracted across the source and target streams is critical yet challenging for the success of unsupervised domain adaptation (UDA). Simply aligning using partial observation of the time series or the local streams leads to poor alignment quality. To address this challenge, we proposed to perform UDA on *the entire data generation mechanism*, which is described by a latent stochastic process conditioned on historical data. This leads to another challenge: aligning two latent stochastic processes tends to result in an intractable discriminator loss. We further derive an analytical upper bound for the discriminator loss to address the second challenge. Calculating such a bound requires evaluating posteriors of the UDA encoder *in continuous time*; this then leads us to formulate the problem using particle filter differential equation (PFDE) rather than traditional Bayesian filtering methods, such as particle filters.
>
> Another advantage of PFDE is that it allows inferring multi-step ahead model distributions before observing the incoming data, therefore effectively reducing the latency (Reviewer jpUT also agreed on this and acknowledged that PFDE is ``"an original contribution"`` ``"with strong motivation for efficient multi-step ahead inference"``). It is worth noting that we adopt *an efficient implementation for solving the PFDE*. Specifically, we adopt surrogate neural networks to approximate the solution. Meanwhile, we use the loss (Eq. (6)) to encourage such solutions to satisfy the governing differential equation of PFDE. Such an approach for solving DEs is commonly adopted by the family of physics-informed neural networks (PINN) [e].
>
> - - -
> **Q4: “ … offline adaptation is biologically plausible… why not for AI tasks?“**
>
> **A4:**  Thank you for sharing these interesting findings in neuroscience. Indeed, offline adaptation is biologically plausible, and it is a smart option when dealing with *stationary* data from the perspective of biological systems. However, several neuroscience studies suggest that internal predictive modeling is crucial for humans performing *real-time* perceptual inferences with *non-stationary* sensory signals [3,4,5,6,7]. This inspires us to incorporate such a mechanism to augment traditional UDA algorithms in handling non-stationary streaming data.

---

> ### Author Response · Authors · 2022-08-02
> **[1/3] Thanks and response to concerns**
>
> Thank you for the careful review and constructive feedback.
>
> Our method contains two key parts: Section 3.4&3.5 on theory of particle filter differential equation (PFDE) and extrapolative continuous-time Bayesian neural networks (ECBNN), and Section 3.6&3.7 on application of ECBNN for UDA with non-stationary streaming data and low latency requirements. We kindly request that Section 3.6&3.7 not be overlooked when evaluating the contribution of our work, as online adaptation with low latency is critical for real-time streaming applications, and existing UDA approaches mostly cannot be directly applied to our problem settings.
> - - -
>
>  Regarding **the reasonableness of the assumption** applied in our approach and the comment “do it in a non-stationary environment (albeit assuming it is stationary which is weird)”, we would like to clarify that:
>
> + We do not assume stationarity for the input streaming data, nor for the continuous-time stochastic dynamical system (describing the time-evolving neural network parameters of UDA progressively receiving non-stationary streaming data) that we aim to infer.
> + We only assume that the marginal distribution of the state $p(\theta(t))$ is a stationary distribution. Note that this is actually a **mild** assumption: the major component characterizing our stochastic dynamical system conditioned on the non-stationary streaming data, namely the posterior distribution of the state $p(\theta(t) | {\bf x}_t )$, still evolve over time, and are therefore **non-stationary**.
> - - -
>
>  As pointed out by the reviewer, the family of neural processes (NPs) [a,b,c,d]
> could be **the potential related work and baselines**. Similar to ECBNN, NPs can infer a stochastic process given partial observations. However, extending NPs to handle our problem settings is highly non-trivial:
> + NPs require taking as input both the input data and the *labels* for *supervised learning* tasks. However, in our *unsupervised* domain adaptation problem setting, the streaming data on the target domain only contains *unlabeled* data during training.
> + Due to domain shift, aligning source and target streams is critical for handling unsupervised domain adaptation settings (to which our setting belongs ). However, how to extend NPs to achieve such alignment is still an open problem.
>
> Besides the aforementioned challenges that prevent us from directly adapting NPs to our problem settings, **ECBNN is distinct from NPs in several aspects which drive several major contributions of this work**:
> + As far as we know, Bayesian treatment of time-evolving neural network parameters for UDA has not been explored by NPs. Reviewer jpUT also agreed and commented that ``"treating the time-evolving neural network parameters as a continuous-time stochastic dynamic system and extending bayesian neural networks to continuous-time domains is a novel approach to unsupervised domain adaptation"``.
> + The stationarity assumed by NPs cound limit their ability to handle the non-stationary streaming data during the real-time testing stage. In contrast, ECBNN adopts a *PFDE* encoding the non-stationary dynamics from the data, thereby continuously and efficiently updating the state of a stochastic dynamical system of neural network parameters as each data sample arrives during real-time testing. Reviewer jpUT also agreed on this and acknowledged that PFDE is ``"an original contribution"`` ``"with strong motivation for efficient multi-step ahead inference"``.
> + We develop an analytical upper bound for achieving temporal-domain-invariant representation learning. Calculating such a bound requires evaluating posteriors in continuous time, while NPs fail to do so.
>
> We will cite these papers and include a similar discussion in the revised paper.
> - - -
>
> Regarding **the concerns about the validity of the proof of Theorem 1**, we would like to clarify that:
>
> + The reviewer mentioned that $p( {\bf x}_t )$ in the proof should not be treated as a constant. Actually, $ {\bf x}_t$ is the observation at time $t$ and $p( {\bf x}_t )$ is the normalizing constant. This can be verified by simply applying Bayes’ theorem for $p(\theta(t)^{(j)} | {\bf x}_t )$.
> + An alternative (maybe more intuitive) way of understanding $p( {\bf x}_t )$ is that: the purpose of the equation is to determine importance weights of each particle $w_t^{(j)}$, where $j\in\{1, 2, \dots, N\}$, at time $t$. Since all $j$’s are associated with the same $ {\bf x}_t$, $p( {\bf x}_t )$ is a constant w.r.t. $j$ and therefore is also a constant w.r.t. $w_t^{(j)}$.

---

### Author Response · Authors · 2022-08-02
**Thanks for the reviews and summary of key paper changes**

We thank the reviewers for their careful and thoughtful reviews.  We are glad that they found our work ”clearly novel” (Reviewer dicg) and  “original” ( Reviewer jpUT) and our experiments  “comprehensive and diverse” (Reviewer jpUT) as well as  “impressive”(Reviewer ZNi1).


Since the concerns voiced by the reviewers are mainly non-overlapping, we offer detailed responses individually to each review.
The key changes to the paper can be summarized as follows:
+ Clarified overview of our method ( Sec. 3.3 in the main paper)
+ Added ablation study (Sec. 7.2.4 in the Supplement)
+ Added timing experiments on the training time of each model (Sec. 7.3.4 in the Supplement)
+ Fixed typos.

---

### Meta-Review · Area_Chair_kzKJ · 2022-08-27

**Recommendation:** Accept
**Confidence:** Certain

**Metareview:**

Paper combines the use of a particle filtering differential equation (newly proposed in this paper) for sampling posterior parameters of a bayesian neural network, with unsupervised domain adaptation, and achieves strong results on tasks demonstrated. Reviewers found the method novel and the problem important. Several questions were raised about the motivation of using Bayesian neural networks, about the utility of combining several of the pieces of the loss function proposed, about the computational cost, but the authors provided satisfactory answers to the questions and provided ablations showing the utility of the different components of the loss function. One downside pointed out was that the method is quite dense and involves several parts that are intertwined. I hope the authors will revise their submission, taking into account the points raised by the reviewers.

**Award:**

No

---

### Decision · Program_Chairs · 2022-09-14

Accept